



# Wind farm flow control: prospects and challenges

Johan Meyers[1], Carlo Bottasso[2], Katherine Dykes[3], Paul Fleming[4], Pieter Gebraad[5], Gregor Giebel[3], Tuhfe Göçmen[3], and Jan-Willem van Wingerden[6]

[1]KU Leuven, Mechanical Engineering, Celestijnenlaan 300A, B3001 Leuven, Belgium
[2]Chair of Wind Energy, Technische Universität München, Boltzmannstr. 15, 85748 Garching b. München, Germany
[3]DTU Wind Energy, Frederiksborgvej 399, DK-4000 Roskilde
[4]National Renewable Energy Laboratory, Boulder, Colorado, US
[5]Siemens Gamesa Renewable Energy, Tonsbakken 16, 2740 Skovlunde, Denmark
[6]Delft University of Technology, Delft Center for Systems and Control, Mekelweg 2, 2628 CD Delft, The Netherlands

**Correspondence:** Johan Meyers (johan.meyers@kuleuven.be)

**Abstract.** Wind farm control has been a topic of research for more than two decades. It has been identified as a core component of grand challenges in wind energy science to support accelerated wind energy deployment and transition to a clean and sustainable energy system for the 21st century. The prospect of collective control of wind turbines in an array to increase energy extraction, reduce loads, improve the balance of systems, reduce operation and maintenance costs, etc. has inspired many

researchers over the years to propose innovative ideas and solutions. However, practical demonstration and commercialization of some of the more advanced concepts has been limited by a wide range of challenges, which include the complex physics of turbulent flows in wind farms and the atmosphere, uncertainties related to predicting load and failure statistics, and the highly multi-disciplinary nature of the overall design optimization problem, among others. In the current work, we aim at providing a comprehensive overview of the state of the art and outstanding challenges, thus identifying the key research areas that could

further enable commercial uptake and success of wind farm control solutions. To this end, we have structured the discussion on challenges and opportunities into four main areas: (1) insight in control flow physics, (2) algorithms and AI, (3) validation and industry implementation, and (4) integrating control with system design (co-design).

## 1 Introduction

Wind farms today consist of tens to hundreds of multi-megawatt turbines working together to provide low-cost energy to

electricity grids across the world. Operating for multiple decades, these machines constantly interact with turbulent flows from the atmosphere, influenced by local orography and further affected by the wakes of upstream turbines. To ensure reliable and low-cost operation and the best possible performance in energy production, control solutions are critical to optimize the turbine power production while balancing loading through the turbine components. Advancements in wind turbine control solutions, collective and individual pitch control, yaw control, and more, have enabled higher energy capture while at the same time

allowing for lighter weight and lower cost machines (Bossanyi, 2003b).

However, operation of wind turbines that are clustered together in a farm is coupled through the flow. Thus, research and development on control for wind energy applications has increasingly shifted to the farm level — where the overall performance,





reliability and cost of the fleet of turbines is considered. Addressing complexities and uncertainties in the physics of wind farm phenomena — from the flow to the machine dynamics to the interaction with the grid — has proved challenging, and ultimately

has limited widespread commercial adoption of more advanced wind farm control solutions. Due to this, wind farm control has been identified as a key component of one of three grand challenges in wind energy science (Veers et al., 2019) that must be resolved to unleash the full potential of wind energy in our future clean and sustainable global energy system.

Core to the challenge of wind farm control are wakes. Wake effects have long been recognized to influence the efficiency of wind turbine clusters, reducing the total energy capture compared to turbines that operate in isolation. With the development

of large wind farms in the 1980s in California, early work looked into the formulation of wake models in wind farms for use in optimization of layout (Lissaman, 1979; Vermeulen and Builtjes, 1981; Jensen, 1983). The first attempt to control wake effects – using axial induction control – was proposed by Steinbuch et al. (1988) with the aim to increase energy capture. Since then, many studies have looked into wind farm flow control, aiming not only at improved energy extraction, but also reduced structural loads, improved balance of the power grid, and combinations thereof (see, e.g., Spruce, 1993; Sorensen et al., 2005,

and many more since then).

The focus of this manuscript is on wind farm flow control (WFFC). We define it as the coordinated control of the turbines in the farm with the aim to influence the flow (wakes, turbulence) in such a way that it improves the overall figure of merit of the farm. The latter can be, e.g., overall power extraction, total lifetime, levelized cost of energy, or simply the lifetime profit. Note that in the literature, the term wind farm or wind-power-plant control includes wind farm flow control, but is often more broadly

used. For instance, early work on wind-plant control focused on grid stability under increased penetration of wind energy, using collective set-point strategies to balance the power grid, lumping all turbines together into one power plant model (Schlueter et al., 1983; Javid et al., 1985; Sørensen et al., 2002; Kristoffersen and Christiansen, 2003). Similarly, the term supervisory control, while sometimes encompassing wind farm flow control, also refers to safe operation, start-up and shut-down, etc., for which flow interaction effects are usually not important. Next to this, the term wind farm control is sometimes also used in the

context of the control of active and reactive power in the local wind farm power grid (Hansen et al., 2002). We refer the reader to these and similar works for broader discussions on wind farm control.

Finally, we note that some earlier reviews and perspective articles on wind farm control already exist. For instance, Kheirabadi and Nagamune (2019) provides a comprehensive review on increasing energy extraction with control. Later, Andersson et al. (2021) and Shapiro et al. (2022) extend the analysis to include provision of grid services objectives, presenting a review of

the corresponding studies. Houck (2022) provides a review of the studies per investigated flow control strategy with respect to power maximization, load alleviation and ancillary services objectives. A recent work of Eguinoa et al. (2021) discusses wind farm flow control in the context of electricity markets and grid integration, providing an initial overview of the capabilities and prospects of the technology from a large-scale systems perspective. Earlier reviews can, e.g., be found in Johnson and Thomas (2009); Knudsen et al. (2015), and Boersma et al. (2017). In this work, we extend past review efforts by focusing on outstanding

research challenges in wind farm control more broadly in terms of (1) insight in control flow physics, (2) algorithms and AI, (3) validation and industry implementation, and (4) integrating control with system design (co-design). A graphical impression of some of the aspects related to wind farm flow control is shown in Figure 1.





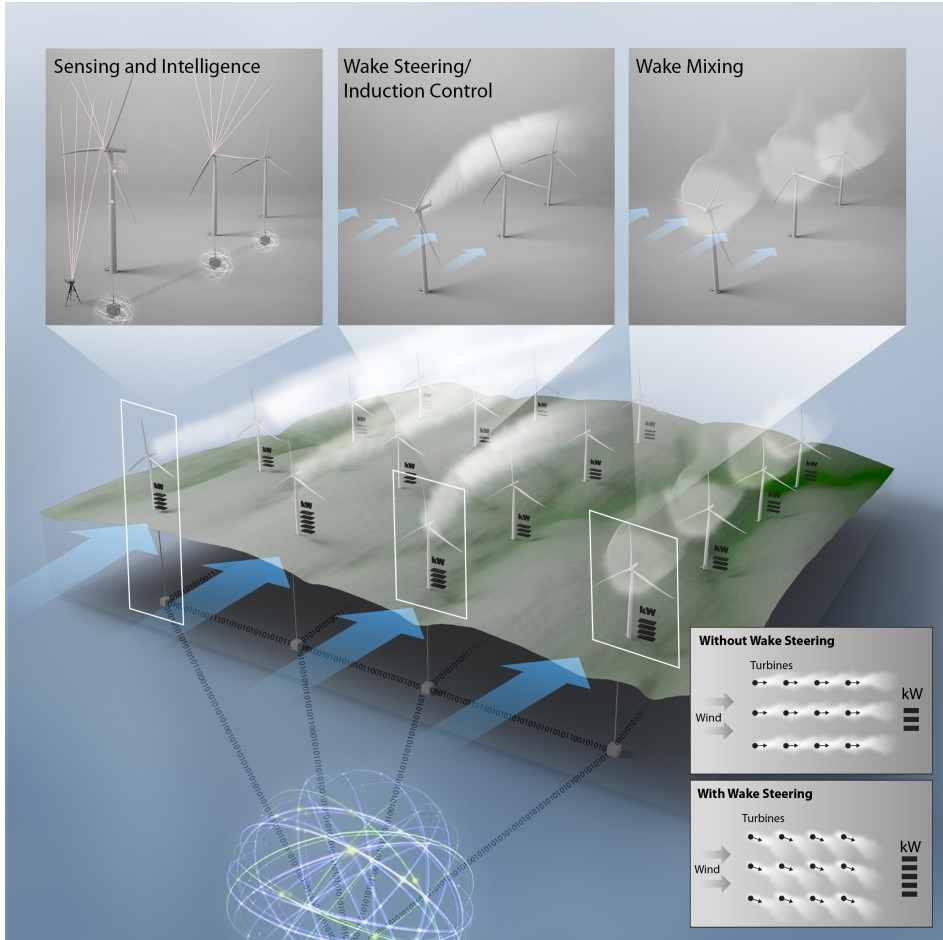

**Figure 1.** Overview of wind farm flow control concepts and important elements that play a role

In the next subsections, we first briefly review the main control objectives that are relevant for wind farm flow control (§1.1), and the main control approaches (§1.2), before we further detail the scope of the current manuscript in §1.3.

## 1.1 Main control objectives studied to date

Very often, the main target of wind farm flow control is to improve the levelised cost of energy (LCoE). This corresponds to sum of all expenses (capital, operation and maintenance (O&M), and end-of-life costs), discounted to a fixed point in time, divided by the power production along the lifetime of the wind power plant (Riva et al., 2019). Beyond improving LCoE and other profitability objectives, wind farm control objectives may include service provision to the electric grid and/or mitigation of adverse social and/or environmental impacts. We briefly review the different control objectives that are commonly considered relevant for wind farm control.



*Increasing energy extraction:* A recent expert elicitation (van Wingerden et al., 2020) involving academic and industrial participants reports that a clear majority of the wind farm (flow) control community considers increased energy production to be the most important benefit of the technology. Its potential value is directly quantifiable for a gain in annual energy production (AEP) and increased revenue through producing more power at a given electricity price. Accordingly, it is the most studied objective of the wind farm flow controllers with several examples of multi-fidelity model implementations, wind tunnel experiments, and field tests reported in the literature. Recent reviews with detailed quantitative comparisons can be found in Andersson et al. (2021), and Kheirabadi and Nagamune (2019). The level of predicted gains varies substantially based on the specifics of a particular case, including the turbine technology, site and resource conditions, and more. Moreover, the confidence in the predicted gains in energy capture depends critically on the physical complexity of the wind farm flow represented in the models used, as well as the accuracy of the measurement sensors and the degree of sophistication of the considered data analysis methods.

*Reducing turbine loading conditions:* The higher levels of turbulence in the wake and effects of asymmetric inflows from partially-waked conditions result in additional structural loading on wake-impinged turbines. This additional loading can affect various components, such as the blades, hub, tower, bearings, transmission, and various actuation systems. Via the reduction of local wake-added turbulence and/or wake redirection, loading on the wind turbines can be mitigated through farm flow control. Compared to the increased energy capture objective, fewer studies investigate dependencies of structural loading on wind turbines to wake control concepts (Kanev et al.; Vali et al., 2019b, 2022). Overall, the potential for load mitigation may be quite significant, and confidence in the predictions may be less sensitive to model fidelity and uncertainty (see, e.g., Campagnolo et al., 2020). However, the capitalisation of load mitigation is harder to assess than changes in the power production, since the connection among loading, component reliability, remaining component lifetime, and ultimately O&M costs is challenging to assess (Clark et al., 2022; Réthoré et al., 2014). Still, some clear benefits may be realised both in the pre-construction/project development phase in terms of wind farm layout and balance of systems costs as well as during the operational phase. Furthermore, the business case for installing load-reducing farm-flow concepts may also be of interest for the realization of lifetime extensions of existing turbines and farms.

*Power tracking for ancillary services and balancing market participation:* As stated earlier, to support the grid stability, a simple approach is to lump the farm together into one power-plant model, and use this to optimize the grid support. Simple set-point distribution schemes, which distribute the farm set point to the different turbines in the farm, are also commonly available (see, e.g., Kristoffersen and Christiansen, 2003; Hansen et al., 2006; van Wingerden et al., 2017). However, for the ancillary services that involve regulation of the active wind farm power output over time spans that surpass the turbine-to-turbine flow time, the ability to control wake interactions may lead to improvements in the way these services are provided. For example, when a reduction in total power output is demanded by the transmission system operators (TSOs), wind farm flow control can be implemented to maximize the (potential) energy extraction and mitigate structural loading, while sustaining the aggregated power level within the quality of the TSO requirements (power tracking) (Ela et al., 2014; Shapiro et al., 2017a; Boersma et al., 2019). Accordingly, flow control can be implemented to maximize the reserve power for higher compensation during mandatory down-regulation (see, e.g., Siniscalchi-Minna et al., 2019), as well as higher value in the balancing market,





and/or minimize fatigue loads at all the selected turbines (see, e.g., Vali et al., 2019b). Similarly, flow control can support better asset management under flexible/dynamic electricity prices (Kölle et al., 2020; Eguinoa et al., 2021). In the zero-subsidy era, maximising the revenue is likely to be prioritised over AEP gains, i.e., reducing the active power during times of low

electricity prices (optimization of loads for potential increase in lifetime instead), up-regulation (boosting) and/or active power maximization during higher electricity prices, maximization of the reserve power for the higher prices in the reserve market, etc.

*Other O&M improvements:* Wind farm flow control concepts can additionally be applied for protection of the power system as well as several turbine components. The former is typically delivered as other types of ancillary services, such as reactive

power and voltage control, and addressed in the broader definition of wind farm control (Hansen et al., 2006). The latter can be in the form of protection against leading edge erosion, icing, overheating of power electronics, etc. In order to mitigate leading edge erosion, typically driven by extreme or aggregated rain events, turbine curtailment via reduction of tip speed is the state-of-the art approach (Bech et al., 2018). Similarly, active pitching is considered to be one of the mitigation techniques for icing on the blades (Sundén and Wu, 2015). For power electronics, avoiding overheating of components can be addressed

through induction control (Ma et al., 2019). While these types of control are focused on the turbine and its components, and turbine–turbine interactions through the flow may not play an important role, these objectives may nevertheless be a relevant contributor to an overall multi-objective wind farm control optimization that includes flow coupling. For example, (leading edge) erosion safe mode via reduced tip speed at the upstream turbines can be combined with the objective of structural load mitigation at the downstream turbines, e.g., through axial induction wake control strategy; potentially adding to the relative

profit increases recently reported by Hasager et al. (2020).

*Mitigation of environmental and/or societal impacts:* Variations in the rotational speed and pitch are commonly adopted as aerodynamic noise reduction techniques at wind farms (see, e.g., Jianu et al., 2012). Additionally, restricted turbine operation, typically in the form of curtailment or turbine shutdown on demand, is used as a mitigation technique for bird and bat collisions, or wildlife fatalities in general (Marques et al., 2014). Again, mitigating control actions are related to turbine control rather

than wind farm control. However, also here, inclusion in a multi-objective framework, either as part of the objective function, or as a constraint, may lead to interesting interactions with modes of operation of wind farm flow control. This can further incentivise the implementation of the technology at certain sites and support the transition towards fully integrated wind farm flow control.

## 1.2 Control approaches

We briefly review current wind farm flow control approaches, in terms of their physical actuation of the wake. The most common mechanisms considered to date include axial induction control and yaw control, and these are the main focus of the current work. However, tilt control is also of interest – particularly for downwind and floating offshore wind turbines and will be briefly discussed. In the future, additional actuators in turbine systems may provide other forms of control, e.g., the use of blade flaps (Barlas and van Kuik, 2007; van Wingerden et al., 2008), but these are excluded from the current discussion.



In reality, wind farm control relies on actuation at the turbine level. This includes control of the torque set-point of the generator, the blade pitch angle, and the turbine yaw setting. However, when discussing wind farm control, very often collective effects on the flow physics that result from turbine actuation are straightaway considered as a control input, without directly considering the precise actuation at the turbine level. The most common example is induction control, in which the axial induction set-point of the turbine is changed to affect the wake and its downstream interactions. This may be achieved in

various ways, i.e. by changing the generator-torque set point (thus changing the rotational speed), the collective blade pitch angles, or combinations thereof. Similarly, yaw control relies on changing the turbine yaw set point, but can at the same time include changes in the generator torque or blade pitch angles. While these types of precise implementation details can matter, e.g., for turbine loading, and should be included in the overall control optimization (see, e.g., §5), from the perspective of the flow they can often be neglected.

Before discussing wind farm control strategies in more detail, we distinguish between two main categories, i.e., *quasi-static* wind farm flow control and *dynamic* wind farm flow control. The former approach changes turbine set points at a relatively slow pace, adapting them to background meteorological variations (change in wind direction, wind speed over the day), but does not react to physical details that happen at time scales that are significantly faster than the farm flow-through time. Dynamic wind farm control aims at including faster flow physics, e.g., accounting for wind gusts traveling through the farm or

for changes of turbine set points traveling through wakes. More advanced dynamic approaches aim at directly influencing the wake mixing and turbulence. We briefly discuss six combinations of strategies that have received a lot of attention in literature: *Static induction control*, *Static yaw control*, *Static yaw and induction control*, *Dynamic induction control*, *Dynamic yaw (and induction) control*, and *Dynamic individual pitch control*. In addition, a short treatment of tilt control (from both a static and dynamic perspective) is introduced. As previously discussed, an increased energy extraction has been the control objective

most commonly investigated so far, but other objectives around loading, reliability, and more have also been considered.

*Static induction control* is the earliest control strategy proposed for wind farm control (Steinbuch et al., 1988). The main idea is to downrate upstream turbines, reducing their axial induction set point and wake strength, in the hope to increase energy extraction of waked turbines. Individual turbine induction set-points can then be optimized at farm level as function of wind direction, wind speed, etc., and used in an open-loop scheme to control the farm. However, although initial results based on

simplified wake models (e.g., Corten and Schaak, 2003; Schepers and van der Pijl, 2007; Horvat et al., 2012; Tian et al., 2014) were very promising, in recent years it was convincingly shown that potential gains in energy extraction are smaller than originally predicted (Gebraad et al., 2015; Bartl and Sætran, 2016; Bartl et al., 2017; Annoni et al., 2016b; van der Hoek et al., 2019). Among other results, wind tunnel experiments conducted with small scale turbines in a neutral boundary layer have shown that, while downrating does energize the wake, it also results in a slower recovery, thereby achieving only modest power

gains downstream (Bottasso and Campagnolo, 2020). Similarly, small gains observed in a recent full scale field experiment Bossanyi and Ruisi (2021) remain within statistical uncertainty. Nevertheless, recently, there is some renewed interest in static induction control, e.g., in tightly spaced farms – see §2.1.1 for further discussion.

*Static yaw control* exploits the fact that wind turbine wakes are redirected when the turbine is misaligned with the incoming wind direction (Clayton and Filby, 1982; Atkinson and Wilson, 1986a), which is a simple result from Newton's action–reaction



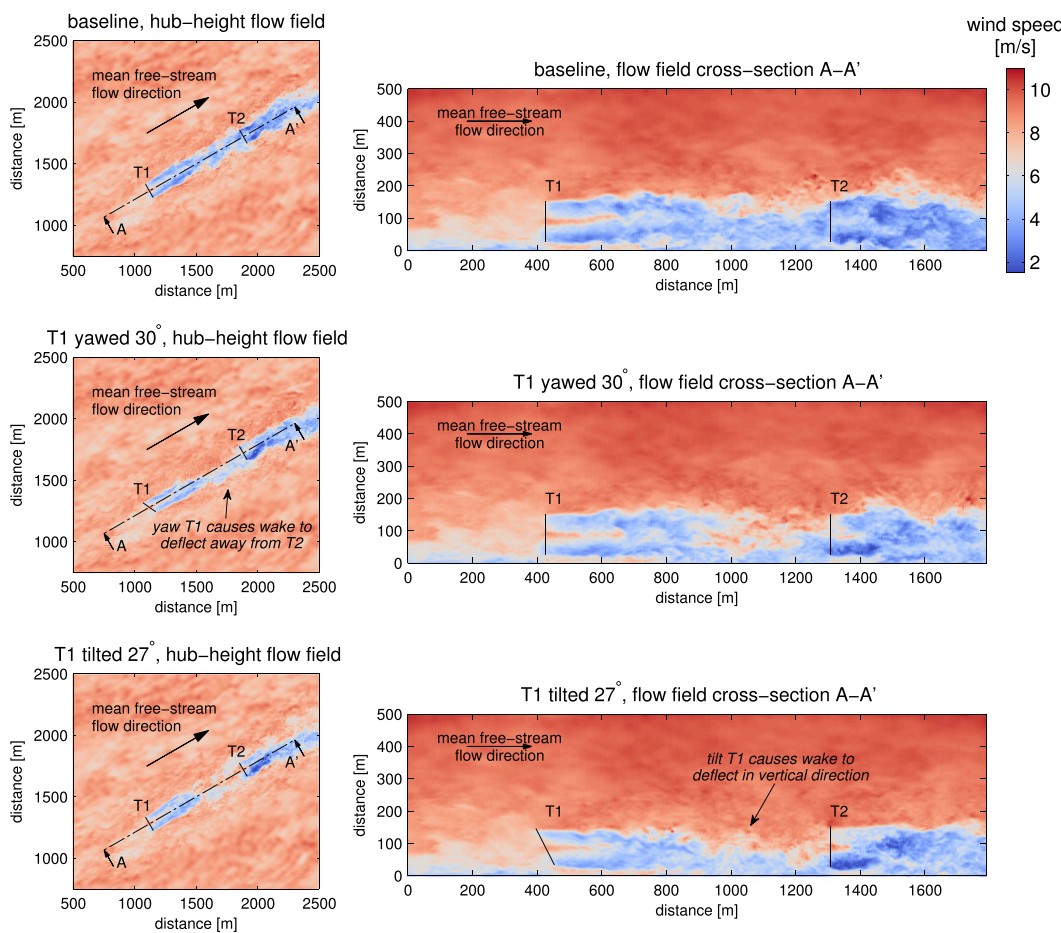

**Figure 2.** Large-eddy simulation comparing yaw and tilt control versus a baseline setup. Figure from Fleming et al., Wind Energy 18, 2135–2143 (2015). Reproduced with permission.

principle. An illustration of the effect observed in LES is shown in Figure 2 (middle panel). Atkinson and Wilson (1986b), and Parkin et al. (2001) were among the first to suggest using this mechanism to influence operational conditions (energy extraction, loads) in downstream turbines. Since then, this approach has received significant attention with strong potential for energy gains observed in simulations (Jiménez et al., 2010; Gebraad et al., 2016), wind tunnels — which provided the first experimental evidence of this method in Campagnolo et al. (2016a), and later in Bastankhah and Porté-Agel (2019), and Campagnolo

et al. (2020) — and in the field (Fleming et al., 2017, 2019; Howland et al., 2019; Ahmad et al., 2019; Fleming et al., 2020; Doekemeijer et al., 2021; Simley et al., 2021). Overall, similar to static induction control, wind turbine yaw set points in a farm can be optimized offline as a function of wind direction, speed, etc., and applied in an open-loop control scheme to the farm. The first commercial products that employ yaw control are currently on the market (Siemens Gamesa Renewable Energy, 2019). Nevertheless, a number of challenges remain, as discussed in §2.1.2.





*Static yaw and induction control* can be potentially combined. In a detailed Large Eddy Simulations (LES)-based optimization study of wind farm controls, Munters and Meyers (2018a) noted that energy extraction by (static) yaw control can be further enhanced by combining it with overinductive induction control (under high thrust coefficient). More recently, the idea was investigated more thoroughly by Cossu (2021) who noted that stronger induction enhances wake deflection while also reducing the losses in the upstream yawed turbine. Pedersen and Larsen (2020) have recently reported that combined yaw and
induction control can increase energy capture and may have secondary benefits in terms of entrainment of additional flow into the plant.

*Dynamic induction control*: In dynamic induction control, turbine thrust set-points are changed much faster with the aim to potentially increase energy extraction over the static situation. Goit and Meyers (2015) and Goit et al. (2016) were first to investigate this approach in an LES-based optimization setting, using a receding horizon control approach, and found theoretical
gains that can be quite high. However, the complexity of LES-based receding horizon control remains too large for practical application. A more practical approach, based on simple sinusoidal variations of the thrust set point was proposed by Munters and Meyers (2018c), and shown to increase wake mixing, thus increasing the energy extraction at downstream turbines. Later, Frederik et al. (2020b) provided a first proof-of-concept in a wind tunnel setting, validating the potential of dynamic induction control. Nevertheless, many issues need further investigation (see §2.2). Finally, we should mention that some studies, using
LES, have also focused on power tracking, dynamically changing individual turbine induction set points (Shapiro et al., 2017a, 2018b). In this case, the dynamic aspect was not so much related to trying to influence turbulent mixing, but rather to dynamically responding to changing wake characteristics due to turbulent wind gusts and changing turbine set points that travel through the wakes.

*Dynamic yaw (and induction) control*: Studies on dynamic yaw (possibly in combination with dynamic induction) are more
scarce. Munters and Meyers (2018a) used LES-based optimization to study the potential of various combinations of yaw and induction control. They found that sinusoidal dynamic yaw control can be used to trigger wake meandering, but that the approach is less effective than static yaw control, and further loses its effectiveness with higher turbulence intensities. They further found when combining dynamic yaw and induction control, that most of the effectiveness comes from the baseline static yaw and overinduction set-points, and that the additional dynamic response to the turbulent background field only brings
smaller improvements. Howland et al. (2020), also using LES, reported similar conclusions for a neutral atmospheric boundary layer. However, results may depend to a large extent on wind farm layout and spacing (Munters and Meyers, 2018b).

*Dynamic individual pitch control*: Cyclic individual pitch control has been used to control turbine loads by reducing the 1P and 3P blade-passing excitation frequencies (Bossanyi, 2003a). The use of cyclic pitch for wind farm flow control was first explored by Fleming et al. (2014) to redirect the wake in lieu of yawing or tilting the rotor (an approach that would technically
resort under 'static' wake redirection), and was then experimentally tested in a wind tunnel by Campagnolo et al. (2016b). More recently, in a LES study, Frederik et al. (2020a) used a modulation version of cyclic pitch control to actuate helical wake modes that improve mixing, and thus power extraction, at downstream turbines. The method has the advantage of not increasing loading as much as dynamic induction control. See further §2.2.



*Tilt control* can be used to provide vertical steering of wakes (see also Figure 2, bottom panel). While most turbines do not include actuators or degrees of freedom that would enable vertical wake steering, such steering may also be achieved for floating turbines using differential ballast control (Nanos et al., 2020). Simulation (Cossu, 2020a, b) studies, and experimental studies with scaled turbines (Scott et al., 2020; Nanos et al., 2020; Bossuyt et al., 2021) indicate that deflecting the wake towards the ground results in larger power boosting than when the wake is deflected upward towards the sky.

## 1.3 Outline

The rest of the paper is organized around four major areas with important fundamental challenges, but also significant potential for improvement. In § 2, we review state of the art and knowledge gaps from the wind farm control perspective in the physics of turbulence, wakes, and the atmospheric boundary layer, as well as what physics can be influenced. Then, in § 3, we explore opportunities for improving algorithms, including state estimation and AI/machine learning. § 4 addresses outstanding needs in experimental validation and demonstration at different scales from simulation to field experiments. Finally, in § 5, we look at opportunities to integrate wind farm control in the broader context of wind turbines and farm design (i.e., co-optimization or co-design of control strategies with the physical system).

## 2 Wind farm flow control physics: turbulence, wakes and the atmospheric boundary layer

In the current section we first discuss challenges and opportunities with respect to quasi-steady flow physics in §2.1. Subsequently, we discuss aspects of wake dynamics and turbulence in §2.2. Finally, in §2.3 the relevance for the control of mesoscale phenomena, such as wind farm blockage, are discussed.

### 2.1 Quasi-steady flow effects

The static wake control concepts rely on affecting the wake through relatively slow changes of the control degrees of freedom of the turbine in a way that affect the time-averaged properties of the wake.

#### 2.1.1 Axial-induction-based control

In the past, most research on axial induction has been on increasing energy extraction. Several studies then demonstrated lower potential for increased energy production from static induction control (see discussion in §1.2). Gebraad et al. (2015) showed that the possible kinetic energy gains in the wake resulting from underinduction are mainly concentrated at the outer part of the wake. Since the wake expands as it flows downstream, this kinetic energy may not be captured by a turbine standing downstream. However, overall gains in power production may still be attainable in situations with partial wake overlap. There are also some indications that underinduction may lead to increased energy extraction in tightly spaced wind farms (spacing less than 4 rotor diameters), where the effect of reducing the initial velocity deficit in the near wake is dominant (van der Hoek et al., 2019). Also full-scale experiments by Duc et al. (2019) show indications that gains may be more significant in a tightly space pair of wind turbines in fully waked conditions. Furthermore, recent efforts have shown some opportunities for increas-





ing power production depending on the implementation of the control strategy (Pedersen and Larsen, 2020). Nevertheless,
demonstrating effective gains in field experiments that are statistically significant remains challenging (Bossanyi and Ruisi,
2021).

The use of overinduction, in contrast to underinduction that is usually considered in axial induction control, is a relatively
new research area. In below-rated regimes, it is possible to reduce the power set-point of a turbine while increasing the turbine
thrust set-point. This can be achieved by increasing the tip-speed ratio, while optionally also adapting the pitch angle (Goit
and Meyers, 2015). Such an operational regime is normally never considered for downrating, as it increases turbine loading.
Overinduction was first considered in a dynamic induction control context by Goit and Meyers (2015) (cf. below). Munters and
Meyers (2016) found, when reducing the response time of dynamic control towards a static approach, that significant power
increases may still be realized using overinduction, whereas this is not the case for underinduction. Martínez-Tossas et al.
(2022) investigated the wake behaviour under overinduction through LES, showing that it leads to faster wake breakup as the
turbine starts to behave more as a bluff body, and derived an empirical model to capture this enhanced recovery mechanism.
Additional research is needed in this area, among others, showing the benefit of this mechanism for wake mitigation and
investigating the effects of increased loading associated with the higher thrust set-points.

Finally, it is well documented that in turbine arrays multiple set-point combinations exist that yield approximately the same
energy output (see, e.g., the wind tunnel experiments of Bartl and Sætran (2016); Bartl et al. (2017)). This can be leveraged
to minimize overall turbine loads while providing power output according to a schedule for the full farm. Recent research in
this area can be found in (Vali et al., 2019b; Baros and Annaswamy, 2019; Galinos et al., 2020; Stock et al., 2020). In order to
successfully apply these concepts, further research and possible test campaigns are needed to validate wind farm load models
in relation to the effect that axial-induction-based wind farm control concepts have on the loads on specific components of
the downstream turbines. This direction of research may be very relevant for LCoE reductions of wind farms in the future, in
particular if included as part of an overall co-optimization framework — see §5 for further discussion.

### 2.1.2 Wake steering using yaw offsets

At present, wake steering through yaw is probably the most advanced control approach it terms of commercial realization (see,
e.g., Siemens Gamesa Renewable Energy, 2019). Nevertheless, a number of research challenges remain in this area, mostly
related to the complex response of the wake to yaw set-points and its strong dependence on atmospheric conditions.
First of all, when the wake is deflected using a yaw offset, it gradually deforms into a curled shape when moving downstream,
deforming into a kidney-like cross section (Howland et al., 2016; Bastankhah and Porté-Agel, 2016; Bartl et al., 2018; Fleming
et al., 2018). As a result of this and the natural wake rotation induced by the turbine torque, the wake deflection is also not
symmetric with respect to the yaw offset (Bastankhah and Porté-Agel, 2016; Bartl et al., 2018; Fleming et al., 2018). However,
it is well known that the wake shape can depend strongly on atmospheric conditions. In particular, stratification (Magnusson
and Smedman, 1994; Chamorro and Porté-Agel, 2010) and wind veer (Abkar and Porté-Agel, 2016; Bromm et al., 2017) are
known to have significant effects on wake behavior. The effect of these parameters in the presence of yaw is much less studied





to date. Also, turbine control can have an effect on the wake of a yawed turbine: e.g., the effect of load-reducing individual pitch control on the behavior of the wake of a wake steering wind turbine was investigated using LES in (Wang et al., 2020a).

Secondly, the structure of a deflected wake includes the presence of a 'transverse wake' (Atkinson and Wilson, 1985, 1986a), i.e., a lateral flow component (or side-wash) that results in the lateral displacement of the streamwise velocity deficit. In turbines that are sufficiently close downstream of a yawed turbine, this effect (called 'secondary steering') generates a change in inflow direction, as first observed in LES simulations by Fleming et al. (2018); Wang et al. (2018) and also confirmed by wind tunnel experiments in the latter reference. Similarly to above, this phenomenon is expected to depend on atmospheric conditions such as stratification and wind veer.

In recent years, significant research has focused on developing wake models that include wake steering, changing wake shape, and possibly also secondary steering (see, e.g., Bastankhah and Porté-Agel, 2016; Shapiro et al., 2018a; Martínez-Tossas et al., 2019; Howland and Dabiri, 2021). Additional parameterization of atmospheric conditions and testing in actual control settings are still largely ongoing research.

Finally, in terms of loading, static yaw control has mixed implications that need to be carefully considered. Yaw control can be used to reduce loading in downstream turbines by slightly yawing the upstream turbine to reduce partial waking of downstream machines. At the same time, however, yaw control that improves energy production may increase partial waking and loading of downstream machines (Herges et al., 2018). Furthermore, the yawed turbine may also see some increase in loading under certain operational conditions (Damiani et al., 2018). Thus, application of static yaw control in practice seeks to balance or avoid negative loading impacts while still achieving benefits in increased power and energy production (Fleming et al., 2019).

## 2.2 Wake dynamics and turbulence

Here we focus on the challenges related to triggering and interacting dynamically with wakes and turbulence as a means to control wind farm flow, and speed up wake breakup, or increase mixing and entrainment into the wakes. Recently, some first studies suggest that this effect may be leveraged to significantly increase energy extraction of wind farms (Goit and Meyers, 2015; Munters and Meyers, 2018c; Frederik et al., 2020a).

### 2.2.1 Wake dynamics

From a phenomenological point of view, two different physical mechanisms that lead to wake breakup have received significant attention: tip/root-vortex instabilities, and wake meandering. Next to this, vortex rings triggered by the dynamic modulation of turbine thrust (Munters and Meyers, 2018c) have also received some attention recently. We briefly discuss these topics here, highlight possible connections to control, and discuss limitations and challenges.

Tip-vortex and root-vortex instabilities have been investigated as an explanation for wake breakup and mixing (Sorensen, 2011; Iungo et al., 2013; Viola et al., 2014; Quaranta et al., 2015). These studies mostly looked at linear stability analysis of single wakes without effectively studying wind farm control. Nevertheless, e.g., Quaranta et al. (2015) suggest that carefully selected modulation of the blade rotation leads to faster wake breakup triggered by tip-vortex instability. However, all of these





studies are at low turbulence intensities for which the wake remains very stable, and dynamics are dominated by the helicoidal
tip-vortex system. At higher turbulence intensities, the tip-vortex system is dissipated relatively fast, and other nonlinear effects
play a dominant role in wake mixing.

A second phenomenon that has been studied extensively is wake meandering (Taylor et al., 1985; Medici and Alfredsson,
2008; Larsen et al., 2008; España et al., 2011; Kang et al., 2014; Foti et al., 2018). This type of instability is mostly triggered
by large-scale motions in the background turbulence and leads to increased dynamic loading on downstream turbines. Munters
and Meyers (2018a) investigated sinusoidally varying yaw control for increased mixing, observing the triggering of wake
meandering. However, the effectiveness of this approach decreased for higher turbulence intensities, as wake meandering in
this case is already triggered (and saturated) by ambient turbulence. Moreover, at low turbulence intensity, static yaw control
yielded better results for the cases studied by Munters and Meyers (2018a) and, although wake meandering will improve power
production, the increased loading may be more of a disadvantage. Nevertheless, whenever wake meandering is present, it is
quite relevant to include it in control models, so that turbines properly respond to time-varying inflow conditions caused by
meandering. To date, wind farm control studies that explicitly include wake meandering in their models remain scarce (Yang
et al., 2015; Munters and Meyers, 2018a; Doekemeijer et al., 2020b).

Munters and Meyers (2018c) proposed a sinusoidally varying thrust coefficient to induce additional mixing in the wake. This
type of actuation leads to the excitation of a train of large annular vortex rings surrounding the wake that entrain high-speed
fluid into the core of the wake (Munters and Meyers, 2018c; Yılmaz and Meyers, 2018) (see also Figure 3). While Munters
and Meyers (2018c) used a simulation setup based on an actuator disk model, Yılmaz and Meyers (2018) investigated the
same type of control using an actuator line model, using a combination of pitch and torque control. Later, dynamic induction
control was also tested in a wind tunnel by Frederik et al. (2020b) and Brown et al. (2021), while a combined actuator-line-
simulation/experimental study addressing both power and loading is reported by Wang et al. (2020d). However, Munters and
Meyers (2018c) did not find evidence that sinusoidal dynamic induction control remains effective when used for other than
first-row turbines, or when used in situations with high turbulence intensities (for which wake mixing is already very high).
When considering a large farm or large turbulence intensities, to date no simple dynamic schemes were identified that can lead
to increased energy extraction (see also §2.2.2 below).

Recently, Frederik et al. (2020a) proposed a different type of dynamic control that is based on a modulation of classical cyclic
pitch control to trigger helicoidal breakup of the wake. A visualization of the typical breakup pattern is shown in Figure 3. The
major advantage of this approach over collective dynamic induction control is that actuation amplitudes are much smaller and
added turbine loading is much lower, while wake breakup may be even slightly better. However, to date, this work has remained
limited to inflow with low turbulence intensity, and a few turbines only. Extension to more complex inflow conditions and large
farms is still under investigation.

The field of dynamic control to trigger faster wake breakup is still very young. Theoretical connections to the field of linear
stability analysis may be of interest, but have not been fully explored. Next to that, methods have only been investigated for
rather simple inflow conditions and small turbine arrays. Effects of stratification, wind shear, and veer, etc. have not yet been
studied.



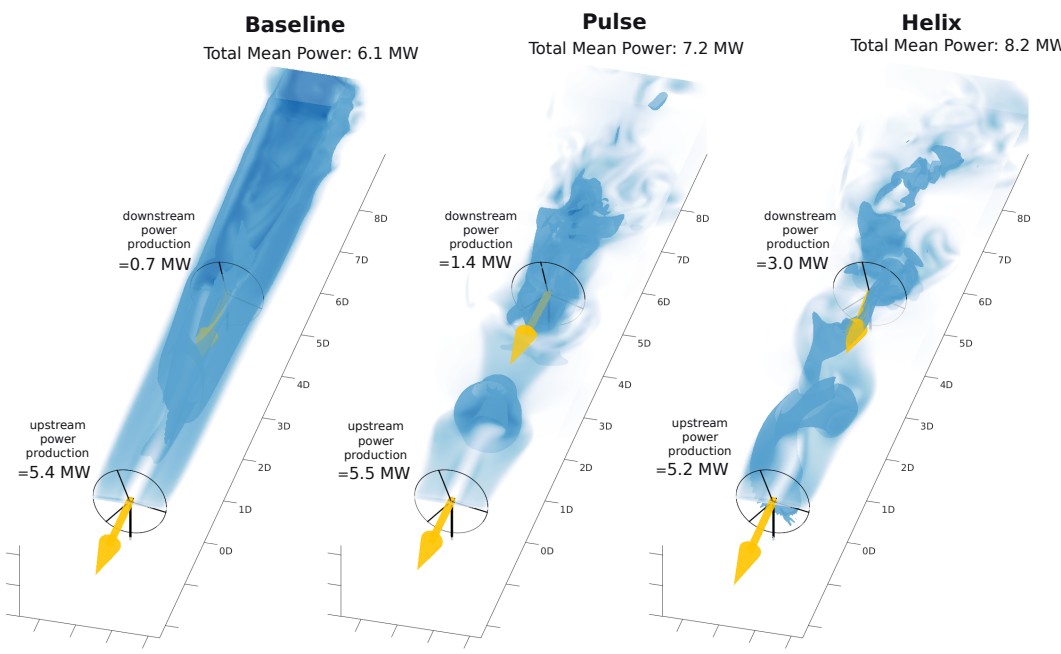

**Figure 3.** Illustration from a LES study of a stable normal operation wake (left), wake resulting from (periodic) dynamic induction control (Munters and Meyers, 2018c) (middle), wake resulting from the helix approach (Frederik et al., 2020a) (right) along downstream distances normalized in turbine diameter (D). Dark blue shading corresponds to an isosurface of the velocity; light blue shading in the horizontal plane corresponds to velocity magnitude. For further details on simulation setup, etc., see Frederik et al. (2020a).

Finally, apart from representing a possible excitation mechanism for active control, wake dynamics may also play an important role in the breakup of wakes when externally forced. For instance, Abraham and Hong (2020) show that dynamic wake modulation by wind gusts or direction changes plays an important role in wake recovery. As another example, floating offshore wind turbines and their movement (heave, surge, roll, and platform yaw and pitch) may be another cause of external excitation of the wake (Wise and Bachynski, 2020) that will gain relevance in the future. Incorporating these effects into improved wake

models for control (e.g., in the context of ancillary service power tracking) is another interesting avenue of future research.

### 2.2.2   Boundary-layer turbulence

The atmospheric boundary layer is the lower part of the atmosphere that is directly influenced by drag and convective heat exchange with the surface. Its height can vary from tens of meters to several kilometers depending on atmospheric conditions, stratification, surface roughness, etc. Depending on the stratification regime, the boundary layer can be very turbulent (unstable

conditions), or nearly laminar and intermittent (very stable conditions). Also, the size of coherent turbulent motions in the boundary layer can change significantly, with a cascade of scales ranging from kilometers down to millimeters in unstable and



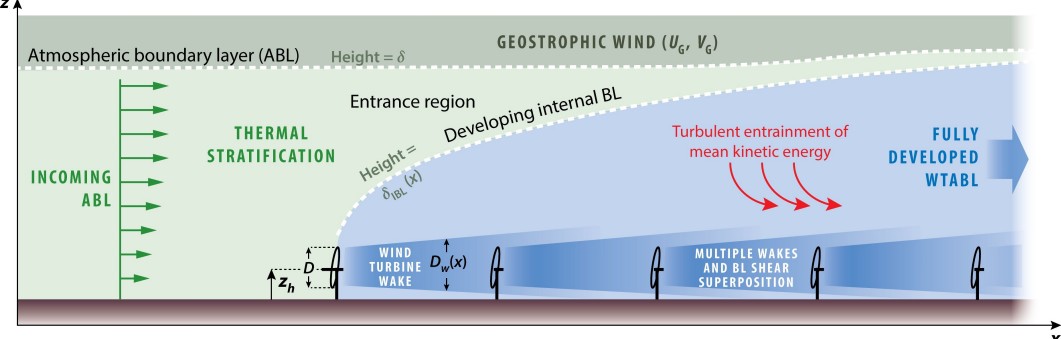

**Figure 4.** Illustration of interaction between the atmospheric boundary layer and large wind farms. From Stevens and Meneveau, Annu. Rev. Fluid Mech. 49, 2017. Reproduced with permission.

neutral conditions, whereas the largest coherent scales in stable conditions tend to be much smaller. For a classical review of the most important aspects that play a role, we refer to Stull (1988) and Garratt (1994).

Given their size, large wind farms interact in complex ways with the atmospheric boundary layer. The size of modern
turbines places them in the middle of the turbulent spectrum (for neutral and unstable conditions), which is important for turbine loading, but also for power variability and interactions of atmospheric turbulence with wakes. The wind farm will also lead to the development of an internal boundary layer, akin to a classical roughness change. An illustration of important interactions that play a role is shown in Figure 4. We refer to Stevens and Meneveau (2017) and Porté-Agel et al. (2020) for recent reviews on the topic of wind farm–boundary-layer interaction.

From a control perspective, the interactions between wind farms and the atmospheric boundary layer may lead to interesting opportunities. The wind turbines can be seen as flow actuators and can be potentially controlled to influence wake mixing, entrainment of high-momentum flow into the farm, etc., possibly leading to increased energy extraction or reduced loading. To date, research in this area is scarce, and has been limited to neutral atmospheric conditions only. Goit and Meyers (2015) were the first to consider wind-farm–boundary-layer interaction in an optimal control setting governed by large-eddy simulations.
Later, Goit et al. (2016) and Munters and Meyers (2017, 2018a, b) followed up on this work, all focusing on increasing energy extraction in the wind farm using dynamic induction control. However, from a practical implementation point of view, this approach is hindered by the large cost and complexity associated with using large-eddy simulations as a control model. Moreover, physical insight in the dynamic controls that can be distilled into simple and robust control laws remains elusive. Munters and Meyers (2018c) performed a detailed analysis of LES-based optimal control to try and identify such laws, uncovering
sinusoidal induction control as a mechanism to increase wake mixing (see §2.2). However, this mechanism only works for first-row turbines, and explains only a fraction of the energy gains observed in the full LES-based dynamic controls. Whether the remaining part of the gain is related to the control of turbulence and entrainment in the atmospheric boundary layer, or rather to the physics of wake mixing in more downstream turbines remains unknown.





Without insights in the control physics, devising a simple controller that leverages wind farm boundary layer interactions
may not be feasible. An alternative would be to build a real controller that directly uses turbulence-resolving simulations, such
as large-eddy simulations, as a control model. Whether it is feasible to arrive at a controller that can be used in real time will
depend on the minimum resolution (Bauweraerts and Meyers, 2019) or possible model simplifications (Boersma et al., 2018)
that still represent the relevant physics while making the control model sufficiently fast. Additional challenges are related to the
correct estimation of the full three-dimensional turbulent state (Bauweraerts and Meyers, 2021) and associated uncertainties
(see also §3). With further advances in computer architecture and flow sensors, such a brute-force optimal control approach
may become feasible in the long term, but will most probably remain out of reach for the next decade.

### 2.3 Mesoscale effects, blockage and wind farm wakes

With wind farms growing in size, effects that extend beyond the scale of the atmospheric boundary layer are becoming visible.
Two effects that have received significant attention are wind farm wakes and blockage. We hypothesize that wind farms may
interact in other ways with mesoscale systems, such as land–sea breeze systems, convective cells, etc. However, to date, these
interactions have not been studied in much detail. Given that wind farms can influence the system on this scale, there may
also be opportunities for control by either mitigating effects (e.g., power loss) or incorporating mesoscale coupling in a more
general optimal control setting.

Wind farm blockage was first observed in field experiments by Bleeg et al. (2018) and they reported possible wind-speed
reductions between 2% and 4% because of wind flowing around the farm (instead of through it). Schneemann et al. (2021)
performed lidar measurements of the Global Tech I offshore wind farm in Germany, showing wind speed reductions up to 6%.
Furthermore, extensive work has studied blockage in simulations and in wind tunnel experiments, with research following two
main working hypotheses for the root cause of blockage. Currently, various new full-scale experiments are underway to further
identify wind farm blockage (see, e.g., Moriarty et al., 2020; RWE Renewables, 2021).
To date, the working hypothesis that has received most attention relates wind farm blockage to classical hydrodynamic
blockage. Segalini and Dahlberg (2020) performed detailed wind tunnel experiments in neutral conditions, showing blockage
effects in the order of 2 to 3%. Earlier wind tunnel studies have shown similar effects for a row of wind turbines, while in
side-by-side arrangements so-called in-field blockage may even increase energy extraction (McTavish et al., 2014). Several
simulation studies have also investigated hydrodynamic blockage, including Meyer Forsting et al. (2017); Wu and Porté-Agel
(2017); Bleeg et al. (2018). When simplifying the flow physics to potential flow and a single axisymmetric open rotor, we
arrive at the Betz limit as a direct expression of hydrodynamic blockage. From this perspective, hydrodynamic blockage in
wind farms would simply be a result of the difference between the expected Betz optimal point for each turbine and the
effective power extraction in a wind farm setup that is not anymore axisymmetric (i.e., when jointly considering all rotor
surfaces). From a controls or optimization perspective, including these additional physics may lead to different operational set-
points (and/or designs) when turbines are clustered together. However, whereas the effects of turbulent mixing of the wake can
be neglected in the Betz theory, it will play a significant role in wind farms, typically reducing wake deficits, and therefore also
blockage. In Lidar experiments, Schneemann et al. (2021) observed that effects of blockage are fully absent in unstable (highly





turbulent) atmospheric conditions, but prominently present in stable atmospheric conditions. Better understanding these effects and including them in control and design models will be an important step to improving wind farm set-point optimization.

A second working hypothesis that has been suggested as an important root cause for blockage is the excitation of atmospheric gravity waves by wind farms. It is well understood that gravity waves are excited in the stably stratified free atmosphere by orography, such as mountains or hills (see, e.g., Teixeira, 2014, for a review). Smith (2010) was the first to suggest that large wind farms may also excite gravity waves, with potential impact on power production because of pressure feedback in the wind farm boundary layer. Allaerts and Meyers (2017) and Allaerts et al. (2018) performed large-eddy simulations of neutral

and stable boundary layers, also showing that gravity waves may reduce wind speeds upstream of the farm, with effects in stable boundary layers being significantly larger. Allaerts and Meyers (2019) proposed a fast engineering model to incorporate the effect of gravity waves in wind farm wake models, partly inspired by a simpler model proposed by Smith (2010). Using this model, Lanzilao and Meyers (2021) performed a set-point optimization study that tried to mitigate energy losses, finding possible power gains of 4% to 14%, depending on atmospheric conditions. To date, experimental data that confirm the existence

of gravity waves excited by wind farms are lacking, and additional research is necessary. Unfortunately, the effects of gravity waves are very difficult to reproduce in wind tunnels, so that more full-scale data are needed. Finally, we note that blockage may be a combination of classical hydrodynamic blockage and wave effects, depending on atmospheric conditions. A first attempt to disentangle both effects in simulations was recently discussed by Centurelli et al. (2021).

    Apart from blockage, wind farm wakes have also received significant attention in recent years. First observations using

satellite imaging already date back to Christiansen and Hasager (2005) for the Nysted and Horns Rev I farm, showing that farm wakes can persist over long distances in stable atmospheric conditions. Since then, several other studies have looked into wind farm wakes (Nygaard and Hansen, 2016; Nygaard and Newcombe, 2018; Platis et al., 2018; Cañadillas et al., 2020) with wakes extending more than 50 km in some cases (Cañadillas et al., 2020). Compared to turbine wakes, wind farm wakes decay much slower. Whether it is possible to improve vertical mixing or redirect farm wakes by control has not yet been established, but

may lead to interesting new research challenges. Whether such mitigation strategies would be economically viable is another open question, in particular since interacting farms are not necessarily owned by the same company.

    Finally, we remark that many more mesoscale effects exist that influence wind farm operation: weather fronts, thunderstorms, coastal gradients, gravity waves originating from orography, etc. However, many of these mesoscale systems are too large for wind farms to be directly affected. In these situations, wind farm control boils down to optimally responding to larger-scale

phenomena, e.g., when they are advected through the farm, while keeping track of flow-feedback effects at smaller scales such as turbine wakes. When considering optimal control, an important challenge in this context is the identification of these large-scale systems, as they are not straightforwardly accessible through local measurements within the wind farm.

## 3   Control algorithms

Understanding the control physics that can potentially be leverage for wind farm flow control (see §2) does not suffice to arrive

at an effective and robust wind-farm controller. At the core of wind farm flow control we have the operational logic/algorithms

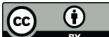

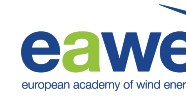


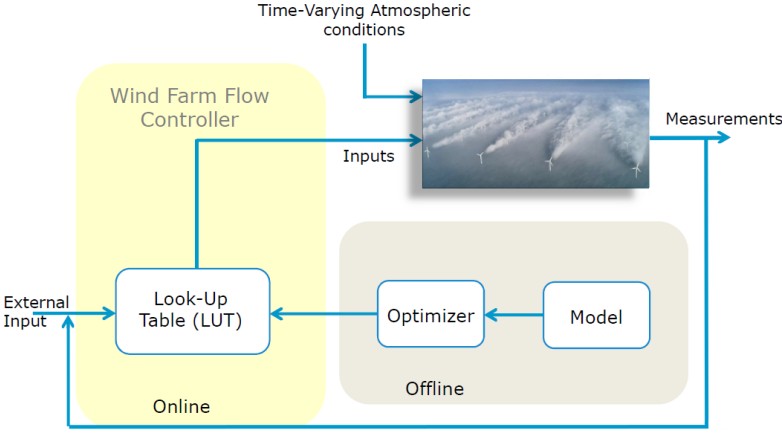

**Figure 5.** State of the art: current practice in wind farm flow control (Horns Rev-I wake picture courtesy: Vattenfall. Photographer is Christian Steiness).

that make the decision on how to operate the individual turbines based on the actual conditions to maximize a certain objective (e.g., power maximization, load distribution or the provision of ancillary services). There are many ways to categorise the algorithms, but in general we can talk about open-loop control and closed-loop control. We start this section by explaining these different concepts and give an overview of the state of the art. Later, in § 3.3 we indicate the synergies with the field of
machine learning and AI. We finalize this section with a summary.

## 3.1 Current practice — Open-loop control

The current baseline for wind farm flow control is defined by many recently performed field experiments in which induction and wake steering are tested (Fleming et al., 2017; Ahmad et al., 2019; Howland et al., 2019; Duc et al., 2019; Fleming et al., 2020; Doekemeijer et al., 2021; Bossanyi and Ruisi, 2021; Simley et al., 2021). In all these experiments the control algorithm
is based on offline calibrated, steady-state engineering models. These models are optimized offline for different atmospheric conditions and look-up-tables (LUTs) are generated which are used online in combination with external inputs (Doekemeijer et al., 2021). These external inputs, such as mean wind direction, wind speed, and possibly other atmospheric conditions such as turbulence intensity, stability, etc., are fed in the LUT to determine the optimal control settings. In traditional control terminology this architecture can be labelled as "open-loop" control and the performance strongly depends on the accuracy of
the model and is prone to disturbances. The information flows as demonstrated in Figure 5 are followed within the "open-loop" control scheme.

However, in academia little attention has been put on the practicality of implementing wind farm control algorithms, as focus has been on proving the essential concepts e.g., maximum power point tracking (Gebraad and van Wingerden, 2015; Gebraad et al., 2013), game theory (Marden et al., 2013; Gebraad et al., 2016), extremum seeking control (Johnson and Fritsch, 2012;
Creaby et al., 2009; Ciri et al., 2017), dynamic programming (Rotea, 2014; Guo et al., 2020). This is especially evident as





few of the efforts consider how to handle sensor uncertainties, actuator uncertainties, and modelling errors and instead focus on maximizing utility under near-perfect conditions. While to date this open-loop approach has been successful in controlled testing environments, it requires human check-ins of small test sites to confirm correct operation (Bossanyi and Ruisi, 2021; Fleming et al., 2020; Howland et al., 2019). To apply wind farm flow control at the scale of whole wind farms and higher temporal resolution and to better manage uncertainties and model errors, robust feedback control is an important feature which can directly accommodate the majority of the inherent uncertainties in wind energy production.

### 3.2 The closed-loop paradigm

For wind farm control the feedback information flows are demonstrated in Figure 6. In this feedback control setting, measurements are used in a real-time optimization framework to determine the next control policy. This framework can deal with, among others, model uncertainty and unknown disturbances. For example, in Doekemeijer et al. (2019) the benefits and challenges for closed-loop control are discussed using the engineering model FLORIS. However, the level of complexity and detail embedded in closed loop control can differ significantly, e.g., steady-state-model-based closed-loop control (acting on quasi-static wake behavior; see §2.1), quasi-dynamic-model-based closed-loop control (including wake delays and responding to meso-scale variability in the flow field; see §2.3), and fully dynamic closed-loop control (predictive control with fully dynamic feedback to the turbulence and updated states; see §2.2). It is evident that the external input, measurement, communication and computational requirements necessary to execute these different types of feed-back control can vary greatly (see also §3.4).

In the foreseen framework real-time data will continuously be used to update a dynamic wind farm model. In combination with an uncertainty description, it will be used to make cautious or robust decisions about the control settings of the individual turbines in a receding horizon framework, by which the turbine's wake can be controlled, minimizing its impact on neighbouring wind turbines for realistic conditions. At the core of this model control paradigm is the fusion of prior knowledge and measurement data (data-assimilation), resulting in data-driven calibrated models that can be used for real-time decision making. These data-driven calibrated models or digital twins can also be used for monitoring wind farms.

The overall challenge is to provide the theoretical foundations, robust integrated designs, and novel optimization routines that will enable the next generation of wind farm flow estimators and controllers. This envisioned framework is at the crossroads of systems and control engineering, optimization, and machine learning and has strong synergies with artificial intelligence. In the remainder of this section we will elaborate on the different building blocks of the closed-loop control paradigm.

*Internal model.* The internal model contains the essential first principles to extrapolate/predict the future behavior of the wind farm as a function of the possible control actions. There are many steady-state models proposed that can serve as an internal model, such as Jensen (Jensen, 1983), Frandsen (Frandsen et al., 2006), Ainslie (Ainslie, 1988), FUGA (Ott and Nielsen, 2014), or Porté-Agel and Bastankah's Gaussian wake formulation (Bastankhah and Porté-Agel, 2016). The main difference among these various methods lies in the characterisation of the steady-state wake, and the wake control effects they include. Several of these models have recently been assembled in open-source wake model libraries such as FLORIS (National Renewable Energy Laboratory, 2021) or PyWake (Energy, 2021). Also in recent years, many dynamic engineering models have been proposed with the capability to predict the dynamic behavior of the wakes as function of a time-varying control input(s).



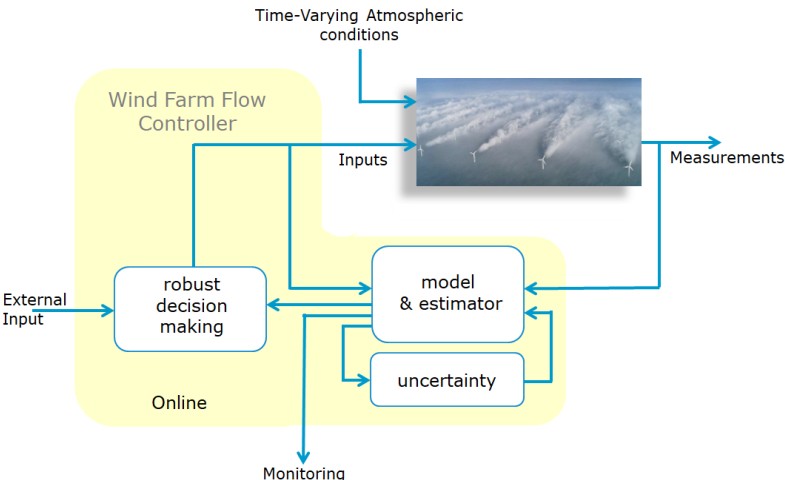

**Figure 6.** Envisioned framework for future wind farm flow control (Horns Rev-I wake picture courtesy: Vattenfall. Photographer is Christian Steiness)

For example, WFSIM (Boersma et al., 2018, 2016; Torres et al., 2011) uses a two-dimensional computational fluid dynamics (CFD) model to predict the dynamic behavior of the different wakes as a function of the time-varying yaw actions. There are many other (quasi-)dynamic models, with different fidelities (representing the flow, possibly the turbines, etc.) that can take a similar role, such as FLORIdyn (Becker et al., 2022), FRED (Van Den Broek and van Wingerden, 2020), PossPOW (Göçmen et al., 2019), FastFarm (Jason Jonkman and Kelsey Shaler, 2021), HAWC2Farm, and the DWM model by Larsen et al. (2008),

etc. Another modelling approach uses data from field measurements and/or high-fidelity CFD to generate low-dimensional surrogate models. This approach, also referred to as data-driven modelling, use techniques such as dynamic mode decomposition (DMD) to generate models (e.g., Liew et al., 2021; Chen et al., 2020; Annoni et al., 2016a; Cassamo and van Wingerden, 2020) and/or machine learning approaches (see also § 3.3). While dynamic or static first principles models easily generalize for different wind direction and speeds these data-driven models face several computational challenges since they need to be

trained on representative data. Schreiber et al. (2020) proposed a hybrid approach that tries to circumvent the limitations of purely model-based and purely data-driven methods, by augmenting a baseline engineering model with correction terms that are learnt from data. Finally, when choosing an internal model, considerations should include: accuracy, computational load, programming language, included/represented physics, disturbance modelling, and fidelity.

*State estimation or data-assimilation, model calibration, model adaptation*. Wind farm flow control depends on the aware-

ness of the current situation of the flow field within the wind farm, the state of the turbines, and the ability to predict based on the current situation/state. In this estimation step all the available measurement data are combined with an internal model to estimate the current situation/state within the wind farm. This is classically called state estimation or data assimilation and recently also referred to as situation awareness. With an updated state the data-augmented model, or digital twin, can be used to predict the future. In control engineering, several types of state estimation techniques such as the family of Kalman




Filters (Doekemeijer et al., 2018; Shapiro et al., 2017b; Doekemeijer et al., 2016, 2017) and direct model inversion techniques (Schreiber et al., 2020; van Der Hoek et al., 2021) are commonly implemented. Before the state-estimation step (as also shown in Doekemeijer et al., 2019), a model calibration step is made, see e.g., Schreiber et al. (2017b) and Doekemeijer et al. (2020a) for the FLORIS model and Sorensen and Nielsen (2006) for the Jensen model. More recently, model adaptation techniques (Andersson et al., 2020; Schreiber et al., 2020) have been proposed to update the first principles model using

data which clearly ties in with the field of machine learning and AI. Besides using these data-augmented models for decision making, it can also be used to provide important information for monitoring purposes since also indirectly measured states can be reconstructed (also labelled as virtual sensors). When choosing an estimation strategy, important aspects that should be considered include: accuracy, sampling time, sensors required, type of model knowledge required, and convergence speed. Depending on the time scales it is expected that different solutions will be optimal.

*Robust decision making.* The (mixed) objective function (e.g., power maximization, load minimization, or power reference tracking) conditioned by the data-calibrated internal model will be used to make decisions on how actuators are employed over time. In general the concept of receding horizon control is used. Based on the current state of the wind farm, the control variables are optimized over a control horizon. The first computed control action will be implemented, and at the next time step a new optimization problem is solved. The optimization problem can be solved with many different algorithms such as

gradient descent and adjoint methods Vali et al. (2019a, 2017).

The important aspects are: reliability, convergence, convexity, adaptability, robustness, ability to work with uncertainty, and computational complexity.

### 3.3   Synergies with artificial intelligence and other digitalisation concepts

With the increased availability and accessibility of the data, the implementation of artificial intelligence (AI) algorithms have

been continuously expanding in the field of wind energy. This is particularly the case for machine learning which is a subset of AI that includes complex statistical techniques (including multilayered neural networks also referred to as deep learning) to enable machines or models to improve at tasks with experience. Machine learning has been increasingly applied to wind farm flow control workflows for the last few years. Currently, the main focus is improving the models and estimators (see Fig. 6) to reduce uncertainties in the predictions, typically without an explicit representation of the external input uncertainties. This

data-informed modelling can include model adaptations via active learning from operational data to correct physical model inadequacies (Schreiber et al., 2020; Andersson and Imsland, 2020) or can be purely data-driven via, e.g., surrogate modelling (Hulsman et al., 2020).

    A central challenge for the machine learning applications within wind farm flow control today is related to the fundamental limitations of the applied algorithms, mainly due to their dependencies on data and computational resources. A first problem

is represented by the insufficient data in terms of its size (e.g., lack of available data), its rate of growth (e.g., delays in data streams, malfunctioning data storage), its variety (e.g., data is accessible only for limited turbine types) and overall uncertainties associated to it. Additionally, observability of all the necessary features that are to be included to predict all the relevant states of the wind farm flow is restrained by the capabilities of the sensors and/or availability of the signals/channels. Addi-





tionally, complex high-dimensional WFFC problems with many features require even more observations to achieve statistical
significance and avoid data sparsity and overfit (generally referred to as the 'curse of dimensionality'; Bellman, 1961).

With broader use of machine learning, the cost of hosting/processing the data and training/updating/inference of the models
is growing rapidly. High performance computing (HPC) plays an increasingly important role in addressing such demands,
enabling their implementation in WFFC to grow significantly. Examples to such growth could be extensive surrogate modelling
based on high-fidelity turbine and flow representation in a wind farm environment (e.g., LES with fully resolved blades; Mittal
et al., 2016), with high spatial and temporal resolution and long(er) simulation time (Andersen et al., 2020). With all the control
hierarchy included, such analyses would be an essential building block for comprehensive digital twins (Grieves, 2014) of the
entire power system (e.g., of a region) to help better advise future planning and operation of renewables.

The knowledge gap in utilising data-driven workflows for a WFFC framework is far greater than physics-based workflows
in terms of the model development and logical sequence of decisions taken to reach a prediction for the stakeholders of
the technology (Feng, 2019). Accordingly, another important challenge for implementing modern machine learning tools in
WFFC is the complexity of the algorithms (Marugán et al., 2018). Typically the models are employed (by domain scientists)
in a black-box manner where the accuracy is prioritised over interpretability or explainability. Although essential, validation in
this perspective is not enough to ensure reliable operation of wind farm(s) as safety-critical systems under strict constraints. A
probabilistic framework and the representation of uncertainty is as crucial for AI approaches as it is for physics-based modelling
for WFFC, as recently shown in, e.g., Rott et al. (2018) and Quick et al. (2020). Transition from deterministic to probabilistic
approaches for AI in wind energy started with Kalman filters (Bossanyi, 1985), and continued for deep learning recently with
Bayesian neural networks (Liu et al., 2020; Mbuvha et al., 2021) and mixture models (Vallejo and Chaer, 2020; Zhang et al.,
2020) capturing the parameter and output uncertainties. Together with other probabilistic architectures, their application to
flow control problems is a growing research interest.

It is also relevant to investigate which features are important in the model predictions and how they are combined (Díaz
et al., 2020). Additionally, decision-level explanations can be provided by breaking down a single or global model into smaller
sub-models and analysing the information flow and series of decisions throughout (Chatterjee and Dethlefs, 2020, 2021). Such
efforts are crucial, as the lack of transparency induces additional risks for large-scale deployment of AI approaches in WFFC.
This prevents the transition from data-supported/data-driven workflows to AI-driven workflows for robust decision making.
The advantage of AI-driven workflows, with AI at the center of the decision-making process, in a WFFC framework is their
capability to process much higher levels of information much faster and to capture very detailed trends and variances. They are
also capable of mapping very nonlinear relationships within the operation to define optima that can be hidden otherwise (see,
e.g., Marden et al., 2013; Ahmad et al., 2016; Zhao et al., 2020). Overall, the AI transformation has a great potential to increase
the efficiency and reliability of the WFFC technology towards the new generation, fully autonomous wind power plants that go
beyond the 'foreseeable' future. It remains to be seen to what extent such transformation will take place and if AI applications
will replace the current model-based WFFC implementation entirely.





### 3.4 Controllability, observability, and sensors

When moving from the current open-loop paradigm to closed-loop wind farm flow control including the use of advanced data-assimilation methods, a number of additional theoretical and practical challenges arise related to controllability and observability of the three-dimensional turbulent flow in the wind farm. Controllability refers to the extent to which flow states and outputs can be influenced by a limited set of control actuators, e.g., the turbine blades in WFFC. From a practical perspective, it is reasonably well documented how the turbine can influence the quasi-steady flow (see §2.1), but much less is known about dynamic control. Although theoretical controllability results exist for various types of systems, (see, e.g., Kalman, 1963; Lin, 1974), these methods become intangible for high-dimensional nonlinear systems, such as wind farm flows.

Observability is the dual of controllability. It identifies how well flow states and inputs can be determined by a limited set of measurements and thus provides an upper bound on the level of flow awareness that can be attained. With more sophisticated control models and control dynamics, more sensors will be necessary to provide statistically relevant state and input estimation. In modern wind farms, sensors usually provide local or point measurements, either located at turbines (including virtual sensors for wind speed, shear, etc. — see below), at meteorological masts, or at the substation. To what extent they suffice as an input to more sophisticated dynamic control algorithms with feedback is still largely unexplored. Nowadays, various remote flow sensing devices (lidar, sodar, radar, drones) promise to improve flow awareness. However, both cost and robustness aspects should be considered when adding more complex sensors into the mix, which should be more than offset by control gains, essentially leading to an integrated cost optimization problem (see §5). Moreover, although extensive work has already been performed on flow reconstruction from, e.g., lidar measurements (see, e.g., Simley et al., 2011; Mikkelsen et al., 2013; Schlipf et al., 2013; Simley et al., 2014a, b; Lundquist et al., 2015; Raach et al., 2017), more sophisticated three-dimensional reconstruction of turbulence still faces many challenges (see, e.g., Lin et al., 2001; Chai et al., 2004; Raach et al., 2014; Bauweraerts and Meyers, 2021).

When considering sensors for WFFC, a lot of them provide flow data directly. These include local sensors such as wind-speed anemometers (cup, ultrasonic), air temperature measurements, wind vanes, and also remote techniques such as lidar, sodar, and radar. However, with advances in monitoring of the turbine itself, indirect flow measurements are also possible that use a model of the wind turbine to invert the structural responses to wind input. These include: the measurement of rotor equivalent wind speed (see, e.g., Soltani et al., 2013); the use of blade-root load sensors (strain gauges) to detect wakes (Bottasso et al., 2018; Bottasso and Schreiber, 2018) or in general to estimate sector-equivalent wind speeds, and hence horizontal and vertical wind shears (Schreiber et al., 2017a) or the use of load harmonic components to estimate shears as well as inflow angles (Bertelè et al., 2021). Given the low cost and relative robustness of the sensors involved, the use of these indirect wind measurements (and their time series) for more advanced reconstruction of the flow field downstream of the rotor, is an interesting avenue of future research.

Finally, as highlighted throughout this manuscript, WFFC is part of a multi-objective optimization problem that also includes turbine loading, cost of O&M, etc. (see also §5). Thus, for proper feedback control, the flow state should be identified, and the turbine state is essential. For instance, using load sensors, loading can be monitored at crucial spots in turbine components by





using digital twin technology, to generate a continuous evaluation of the accumulated fatigue damage (Pimenta et al., 2020). This type of turbine self-awareness requires a well-considered sensing strategy at the level of the turbine and its controller. A further discussion of these elements is, however, outside the scope of the current paper.

## 4 Validation and industrial implementation

The final ambition of research in wind farm control is commercial implementation. However, next to a commercial cost-benefit analysis and various other aspects (as for example certification; see Manjock et al., 2020), it is necessary to develop a thorough validation and proof-of-concept of each new control idea and of its implementation. A recent expert elicitation on wind farm control identified validation as the key priority for advancing wind farm control (van Wingerden et al., 2020).

Several wind farm control concepts have been shown in the past few years to have the potential to increase wind farm pro-
duction. Some concepts have made it to commercial application: for example, wake steering is now available as a commercial product (Siemens Gamesa Renewable Energy, 2019). However, the realization of such commercial opportunities is invariably supported by comprehensive research and development programs across many organizations and countries over several years.

A wind farm represents a high-value asset in which many parties are involved (one or more wind farm owners, a wind farm operator, an original equipment manufacturer, and possibly insurers, financiers and certification bodies). One or more
stakeholders also share in the project revenue generation, as well as in the costs and risks. Thus, demonstration should not only address the potential of a new innovation (such as a wind farm flow control strategy), but also the quantification of the risks of that innovation that are critical to industrial adoption. Such demonstrations must be grounded in well-designed simulation experiments and test campaigns, which generate data and results that are realistic, transparent, and reliable.

In recent years this has led towards a four-stage approach to developing confidence and trust in control concepts, ordered
in increasing cost and effort: (1) proof-of-concept in a sufficiently advanced high-fidelity simulation environment, such as large-eddy simulations; (2) validation of the control concepts in a (boundary layer) wind tunnel; (3) field campaign on research turbines or older assets with a limited scope (e.g., involving a small number of turbines); and (4) full-scale field campaign. Initially, a high-fidelity simulation or wind tunnel experiment is a key enabler for further R&D investment decisions, and for further effort being put into testing at full scale. In recent years, the sequence of simulation to scaled testing to field
demonstration has proven to be a model for commercialization of wind farm flow control solutions. However, each of these stages faces specific challenges with significant room for improvement. These are further addressed below in detail in §4.1–§4.3. Finally, additional aspects with respect to full-scale industrial implementation are discussed in §4.4.

### 4.1 Proof-of-concept studies in high-fidelity simulation tools

In recent years, LES has emerged as a virtual environment capable of representing significant features of wind farm flows
(see, e.g., Calaf et al., 2010; Porté-Agel et al., 2011; Churchfield et al., 2012; Wu and Porté-Agel, 2013; Fleming et al., 2014; Witha et al., 2014; Wang et al., 2017; Martínez-Tossas et al., 2018), making it an ideal choice for the testing of wind farm control strategies (Goit and Meyers, 2015; Munters and Meyers, 2018c; Vali et al., 2019b; Wang et al., 2019; Doekemeijer





et al., 2020b). The advantages of wind farm LES include a high degree of control over the simulation and the ability to run counter-factual simulations. For instance, running a case with and without controls while using the exact same inflow provides a very clear means to compare performance, something that is not possible in full-scale field experiments. Additionally, the full flow field can be readily accessed and analyzed in space and time, contrary to the experimental scaled and full-scale cases, where synchronous and high-resolution measurements of the complete field are not (yet) possible. High-fidelity CFD simulation tools have become more widely available and are used by an increasing number of research teams (for example, LES codes like SOWFA by NREL, SP-Wind by KU Leuven, EllipSys3D LES by DTU, PALM by Leibniz University Hannover, and TUM.LES by TU Munich, which has specialized in the CFD replication of wind tunnel experiments). Nevertheless, many challenges and opportunities remain.

First of all, LES of wind farms are computationally expensive, usually requiring high-performance computing. Simplifying assumptions are often necessary to keep computational resources manageable. Although fully blade-resolved simulations of wind turbines are becoming feasible (see, e.g., Kirby et al., 2019; Sprague et al., 2020), these simulations are currently too expensive to perform over the sufficiently large spatial and temporal domains that are necessary for wind farm control testing. Therefore, a major simplification that is usually made is related to the turbine representation, which is either based on actuator disk (Srensen et al., 1998; Jiménez et al., 2007), actuator line (Sørensen and Shen, 2002; Troldborg et al., 2007), or actuator sector models (Storey et al., 2015; Vitsas and Meyers, 2016). For wind farm control proof-of-concept studies, usually an actuator line or sector model, coupled to an aeroelastic turbine representation and its controller is preferred, allowing not only the evaluation of power output but also of turbine loading. However, the accurate representation of some flow characteristics remains a challenge, including the correct representation of dynamic features related to the blade boundary layer (dynamic stall, rotational augmentation, three-dimensional effects), as well as near-wake features related to the tip-vortex system (Martínez-Tossas et al., 2017; Meyer Forsting et al., 2019).

Secondly, a representative parametrization of inflow conditions in LES remains an important area of research, with a large potential impact on the quality of the results. One aspect of particular importance is the generation of realistic inflow turbulence. In the past, research has focused on the development of synthetic turbulence approaches (see, e.g., Wu, 2017, for a review), but nowadays it is well established that precursor methods yield more realistic spatio-temporal correlations in the flow (Stevens et al., 2014; Munters et al., 2016). Another aspect that needs attention is that simulations are often performed using idealized background conditions, whereas the atmospheric inflow is governed by mesoscale weather systems that usually display a complex amalgamation of larger scale flow features. In recent years, research has focused on coupling mesoscale models to microscale LES (Muñoz-Esparza et al., 2014; Muñoz-Esparza and Kosović, 2018; Haupt et al., 2020), but issues remain with respect to the characterization of representative turbulence at the LES inflow, which is not directly represented in mesoscale models.

Finally, although LES is a quite mature simulation technology that has been developed for more than three decades, some open problems remain with respect to subgrid-scale modelling, in particular in stably stratified conditions (see, e.g., Couvreux et al., 2020), and with respect to wall-stress models (required for the simulations of the atmospheric boundary layer) and the representation of complex terrains.





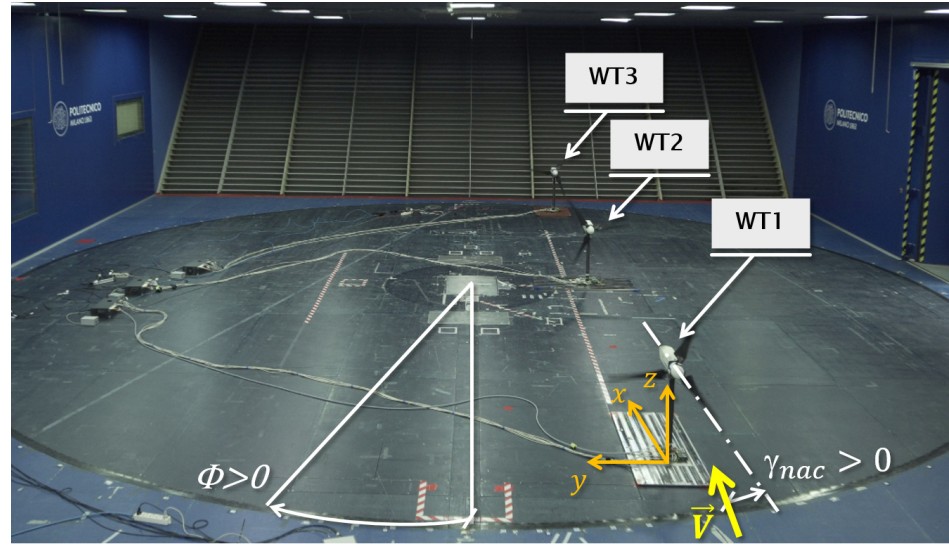

**Figure 7.** Wind tunnel facility of the Politecnico di Milano with a three-turbine setup for testing yaw control. Picture reproduced from Campagnolo et al. (2020), Wind Energ. Sci., 5, 1273–1295. Reproduced with permission.

## 4.2 Validation in wind tunnel experiments

Wind tunnel testing with scaled models has long supported scientific research in various engineering fields, and wind energy is no exception. In addition to the classical characterization of the aerodynamic performance of airfoils (which, for wind energy applications, poses its own challenges due to the combination of high Reynolds numbers and unsteady inflows), wind tunnel testing has recently moved to the simulation of clusters of scaled wind turbine models exposed to flows that mimic the characteristics of the atmospheric boundary layer. Working hand in hand with simulation and full-scale experiments, wind tunnel testing contributes to the understanding of wake physics and the validation of wake models (Chamorro and Porté-Agel, 2009, 2010; Hu et al., 2012; Iungo et al., 2013; Viola et al., 2014; Bastankhah and Porté-Agel, 2015; Howard et al., 2015; Yang et al., 2016; Bastankhah and Porté-Agel, 2017b; Schreiber et al., 2017b) and, in turn, to the development of wind farm control technology (Campagnolo et al., 2016a; Bastankhah and Porté-Agel, 2016; Wang et al., 2017; Campagnolo et al., 2020; Frederik et al., 2020b; Wang et al., 2020d). Although it is clear that any wind farm control method has to be demonstrated in the field before it can achieve full credibility, wind tunnel testing of control concepts has been gaining an increasing popularity in recent years for proof-of-concept demonstration.

The success of wind tunnel testing is due to some of its distinguishing characteristics. First, differently from the field, the ambient conditions in a wind tunnel are precisely measurable, repeatable and —at least to some extent— controllable. A number of boundary layer wind tunnels worldwide are capable of generating realistic scaled turbulent flows. Among these, the large facility at Politecnico di Milano, which features a 3.84 m (height) by 13.84 m (width) by 36 m (length) test section, has been used for some of the most complex wind farm control experiments to date (Campagnolo et al., 2016a; Bottasso



and Campagnolo, 2020; Campagnolo et al., 2020; Frederik et al., 2020b; Wang et al., 2020d). In a wake steering experiment, dynamic changes in wind directions have been simulated in this facility using its large actuated turntable (Campagnolo et al., 2020) — see Figure 7. Apart from size, which is important to limit blockage and avoid excessive miniaturization of the models (Bottasso and Campagnolo, 2020), some facilities have been designed to generate specific flow conditions that are particularly

relevant to wind energy applications. For example, the wind tunnels at the University of Surrey (Hancock et al., 2014) and at the University of Minnesota (Chamorro and Porté-Agel, 2010) can generate stratified flows, such that not only neutral but also stable and unstable boundary layers can be investigated; the WindLab wind tunnel (University of Oldenburg, DE) features a large active grid (Neuhaus et al., 2021), which gives more freedom in the generation of turbulent flows than the classical passive means (Zasso et al., 2005); while the WindEEE facility (Western University, London, Ontario, CA) (WindEEE, 2020)

can generate tornadoes and down-bursts, which are relevant for the simulation of extreme operating conditions.

A second distinguishing feature of wind tunnel testing is that measuring in a lab environment is usually easier, more accurate, and cheaper than in the field. In fact, detailed characterizations of the inflow and of the wakes can be obtained with a variety of techniques and equipment that differ in cost, accuracy, spatial and temporal resolution, intrusiveness, and setup complexity. These include pressure and hot-wire probes (Lomas, 1986), particle image velocimetry (PIV) and its derived techniques

(Adrian, 2005), and also —more recently— scanning lidars (van Dooren et al., 2017). Finally, when properly scaling a set-up to smaller spatial dimension while keeping velocity roughly at the same magnitude (as typically done for wind tunnel testing), also the time scales will decrease (time flows faster)(Bottasso and Campagnolo, 2020; Canet et al., 2021; Campagnolo et al., 2020), so that more statistical information can be accumulated over shorter experimentation times.

A third useful characteristic of scaled testing is that models can be designed ad hoc to achieve specific goals (Schottler et al.,

2016; Lanfazame et al., 2016; Winslow et al., 2018; Kelley et al.; Bastankhah and Porté-Agel, 2017a, b; Campagnolo et al., 2016a; Bottasso and Campagnolo, 2020; Nanos et al., 2021), and can be extensively instrumented (Bottasso and Campagnolo, 2020). Advances in 3D printing (Zhu, 2019), improvements in precision manufacturing and in the realization of complex composite structures (Campagnolo et al., 2014; Bottasso et al., 2014; Parandoush and Lin, 2017), advanced sensors (Qiu et al., 2020), miniaturized actuators, wireless transmission, and other technological improvements are being actively exploited to

improve model building (Bottasso and Campagnolo, 2020). While the small dimensions often pose constraints, the flexibility offered by the design of scaled turbine models is hard to match when compared to full-scale machines. Sophisticated lab characterizations of crucial model components are also possible (Campagnolo et al., 2014; Wang et al., 2020c). Furthermore, farm layouts and scenarios can be readily changed to explore different conditions of interest by simply repositioning the scaled model turbines on the wind tunnel floor.

In parallel to these useful characteristics of wind tunnel testing, it is important to realize that in general a scaled model cannot exactly represent all physical processes that take place at full scale. Dimensional analysis, through the Buckingham Π theorem (Buckingham, 1914), lays the foundations for the systematic development of scaling laws for the design of model turbines. Depending on the application and focus of the research, scaled designs can differ significantly. For instance, in a wind tunnel of the same size, a focus on rotor aerodynamics may lead to a larger rotor design than when a study of wake behaviour is the sole

focus. Other aspects that can have a decisive impact on design are a possible focus on reproducing correct (scaled) aeroelastic



behavior (Bottasso et al., 2014), or the inclusion of realistic turbine actuators. Moreover, the wind tunnel facility that is used will impose some additional constraints, so that the design of a scaled model is an exercise in finding a best compromise, as it is often the case in many complex engineering applications.

Canet et al. (2021) reviewed the laws that govern steady and transient gravo-aeroelastic scaling of wind turbine rotors, while Bottasso and Campagnolo (2020) applied the results of the scaling analysis to the design of wind turbine models for wind tunnel testing; Wang et al. (2020b) looked specifically at the realism of the wakes generated by scaled models for wind farm control applications. The main findings of these studies show that, notwithstanding the existence of some mismatched effects mostly caused by the chord-based Reynolds number, with proper design choices scaled wakes generated in wind tunnel experiments appear to be remarkably similar to their full-scale counterparts, except in the immediate proximity of the rotor.

A final aspect that is important when considering scaling, is that a reduction of length scales leads to a reduction of time scales, i.e., an acceleration of time in the scaled experiment (Bottasso and Campagnolo, 2020; Canet et al., 2021; Campagnolo et al., 2020). Thus, control algorithms and actuations need to be executed much faster in a wind tunnel than at full scale. This poses constraints on the turbine hardware, in terms of actuation rates, and on the computing hardware and software, in terms of execution speed (Campagnolo et al., 2020). Therefore, depending on the setup, the testing of complex control algorithms 755 (e.g., model-predictive control) may not be feasible, or may imply considerable challenges.

Notwithstanding the indisputable success of wind tunnel testing in recent years, much remains to be done on many fronts. For example, PIV and lidars should become more widely available, because accurate nonintrusive flow measurements are indispensable for better understanding of the flow physics, as well as for model validation. The effects of floating motions on wake behavior and control are not well understood; however, the scaled testing of floating wind turbines is still in its 760 infancy because of the challenge of replicating at scale (or by hardware-in-the-loop co-simulation) aero and hydrodynamic interactions. Improvements are also necessary to more faithfully replicate at scale the inflows that very large modern machines face in various types of atmospheric conditions, together with the effects that terrain orography and vegetation cause on the flow and on wake behavior. The development of complete high-fidelity digital simulation-based replicas of experimental setups —including the wind tunnel, the turbine models, and the turbine- and farm-level control laws— can greatly increase the value 765 of experimental measurements (Wang et al., 2019).

### 4.3 Validation via field tests

Full-scale validations implement wind farm controls on industrial wind turbines and seek to validate wind farm control models and performance. Some example studies at this scale include tests at the Summerview wind farm (Alberta, Canada) (Howland et al., 2019), the Peetz Table Wind Energy Center (Colorado, United States) (Fleming et al., 2019, 2020), the Sole du Moulin 770 Vieux farm (Ablaincourt-Pressoir, France) (Ahmad et al., 2019), and the Sedini wind farm (Sedini, Italy) (Doekemeijer et al., 2021). The advantage of these tests are that they represent the realized concept in near final form, with full-sized turbines in fully realistic atmospheric conditions. However, the challenge is the much reduced controllability and observability of inflow conditions. Since identical conditions can not be guaranteed to be applied to the plant with and without wind farm controls, statistical methods must be used to try to quantify changes in performance. A common approach, for example in Fleming et al.





(2020) and Doekemeijer et al. (2021), is to test the wind farm controller on only some of the turbines in the farm (putting all other turbines in a reference group) and then toggle on and off the control group, producing a baseline and test data set. By comparing the relative performance of turbines being controlled to the reference turbines (through power ratio or energy ratio comparisons, for example) when the turbine is off versus on, the effect on power or energy production can be estimated. Toggle testing has been an established practice in testing new controllers for wind turbines (see Bossanyi et al., 2013). However,

wind farm wake control strategies, unlike turbine control strategies, have a dependence on wind direction that complicates the comparison. Additionally, previous tests of turbine control strategies, based on pitch or torque control, could toggle quickly, making 10-minute intervals a standard choice. However, in wake control, the yaw control changes much more slowly, and the changes in the wake need time to propagate downstream. These influences have made approximately 1-hour toggling typical of early tests. The establishment of best practices in performing field validation is actively researched and a focus point of the

new IEA Wind Task 44: Wind Farm Flow Control (https://iea-wind.org/task44/).

To date, there are many opportunities for advances in full-scale validation. First, collecting data in field campaigns using more turbines within farms, or even whole farms, and for longer periods than currently reported, will provide much more certainty over performance. A next opportunity is to employ more sophisticated measurements to learn about the fundamental properties and performance of flow control (for example, lidars, radars, drones and etc.).

There are also opportunities for testing more sophisticated control models. To date, many of the wind farm controllers tested in public field trials of wake steering are based primarily on the open-loop paradigm and precomputed look-up tables. Future field tests should incorporate more recently developed control strategies such as consensus for collectively identifying wind directions (Annoni et al., 2019) or use online estimation (Doekemeijer et al., 2020b) or robust design (Rott et al., 2018). Further, consideration of the underlying turbine yaw control optimized to implement wind farm control is an important area of ongoing

work (Kanev, 2020).

Finally, there are many opportunities to measure and quantify impacts on loads, in addition to confirming energy uplifts. While there is some research in the area of field trials of wind farm controls measuring impact on loads (Damiani et al., 2018), this is another critical area of ongoing research.

## 4.4 Industrial implementation

Once a wake control concept has been proven, several steps need to be taken in industry to achieve commercial application in operating wind farms. This is illustrated in Table 1, where we suggest definitions of technology readiness level (TRL) for wind farm control concepts from the identification of basic principles (TRL 1) to commercial application (TRL 9). Challenges in transitioning from TRL 5 (first field test) to commercial implementation TRL 9 are further discussed in this section.

First, the effects of wake control need to be shown to be reproducible and measurable from sensor data that is readily

available on a wind turbine rather than only seen in an experimental setup with special remote sensing equipment. This ensures that the performance of a wind farm control concept can be shown to be working at several operating wind farms and that it is therefore scalable as a product. This step often requires adjustment of the available measurement equipment on commercial wind turbines. The adjustments may include calibration of the measurements for the alternative operational modes needed for





the flow control, which is a requirement that could already be considered early on in the formulation of research questions and the setup of simulations or wind tunnel tests. Further, a typical practical issue to be solved is the accurate calibration of wind direction or nacelle yaw position relative to North at each site, which makes sure that wake control is done in the appropriate wind directions. Secondly, if a flow control mode is expected to increase certain loads acting on a wind turbine, an evaluation needs to be performed that shows whether the loads are still within the envelope that the turbine is designed for. Moreover, special care needs to be taken in the evaluation of actuators if they are used in a different way than originally designed for. For new wind turbine models, the design load cases could be adjusted to allow for more aggressive wind farm control strategies. This requires an integrated co-optimization of design and control. See § 5 for further discussion. Thirdly, the park-level control and turbine controller needs to be implemented with relevant safety features. This includes cybersecurity measures allowing safe communication between the wind turbine and wind park level control.

The three steps mentioned above allow transition from a field test (TRL 5) to safe implementation of the park-level control on a wind park without special human supervision (TRL 8). For transitioning to TRL 9, commercial application, not only does the control implementation need to be deployable on several sites in order to be able to scale the product, but also the commercial value of the control needs to be proven. If the objective of the wind farm control is to improve the wind farm AEP, this may entail validating model predictions using data from one or more farms, and then using the model to project a longer-term benefit for specific sites. Alternatively, a more data-driven, site-specific approach can be used where the AEP benefit is measured from a toggle test at a specific site in order to measure performance of the wind farm control on that site. Such a toggle test may take several months to measure the park control benefit in a large range of wind conditions, or even up to a year to take into account seasonal variations. An example approach for such a toggle test was presented in Boccolini et al. (2021). The wind industry further needs to establish relevant certification and standards related to the above-mentioned topics of loads evaluation, safety, and wind park production benefit prediction and measurement. A position paper on certification practices for wind farm control was recently presented by Manjock et al. (2020).



**Table 1.** Definition of Technology Readiness Levels (TRLs) for wind farm control concepts, and example references.

| TRL | General definition | Specific definition for wind farm control concept | Example for yaw-based wake steering |
|---|---|---|---|
| 1 | Basic principles observed | Identification and understanding of aerodynamic effect in a wind park | Wake deflects when yaw offset applied (Clayton and Filby, 1982) |
| 2 | Technology concept formulated | Understanding how certain aerodynamic effects could be used/manipulated/controlled to the benefit of wind park performance | Yaw offsets can be controlled strategically to steer a wake away from downstream turbines in a way that improves power production of wind park. (Atkinson and Wilson, 1986b; Medici and Dahlberg, 2003) |
| 3 | Experimental proof of concept | Proof of concept in simple setup (for example two turbines) in wind tunnel or high-fidelity simulation | Wake steering shown in LES to improve power production for two turbines in (Jiménez et al., 2010; Fleming et al., 2015) |
| 4 | Technology validated in lab | Control demonstrated in wind tunnel test and/or high-fidelity simulation with wind park setups | LES study with wake steering park controller for 6 turbine setup (Gebraad et al., 2016) and wind tunnel experiments with wind turbine scaled models and a wake steering park controller (Campagnolo et al., 2016a), both demonstrating production increase |
| 5 | Technology validated in relevant environment | Demonstration of control in field test with several wind turbines | Field campaign evaluating wake steering controller at a wind farm (Howland et al., 2019; Fleming et al., 2019). Production increase demonstrated for two closely spaced turbines. |
| 6 | Technology pilot demonstrated in relevant environment | Experimental control software demonstrated on wind farm, working in limited range of wind conditions. | Wake steering demonstrated on a wind farm for particular selected three-turbine arrays and wind direction sectors (Doekemeijer et al., 2021) |
| 7 | System prototype demonstration in operational environment | Experimental control software demonstrated on full-scale wind farm, working in full range of appropriate wind conditions. | Wind park wake steering controller prototype working on full wind farm. Wake steering active in below-rated wind speeds and in all wind directions where there are wake interactions between turbines |
| 8 | System complete and qualified | Park-level control and turbine controller implementation with relevant safety features and certification based on loads evaluation. | Implementation of park-level control automatically commanding yaw offsets in relevant wind conditions. Turbine software augmented yaw offsets. Approved loads evaluation of yaw-offset operation |
| 9 | System proven in operation | Commercial application of wind farm controls product in wind parks | Commercial application of wake steering control in wind farms (see e.g., Siemens Gamesa Renewable Energy, 2019) |





It is also important to note that the advancement through the TRL stages also depends on specific technologies and site characteristics. Thus, a given control strategy may advance at different rates for different applications. For example, whereas wake steering is advanced to TRL 9 for land-based and fixed-bottom offshore wind farms, this is currently not the case for floating wind energy applications. In addition, on land, conditions related to complex terrain or other features of the wind farm design have yet to be fully validated from a yaw-based wake steering perspective. The variation of performance and risks of a control strategy for different technologies, site design, and site environmental conditions require ongoing R&D well beyond the first instance of commercialization.

So far, industry has shown more interest in wind farm control concepts that result in AEP increase, but in a subsidy-free market more interest may go to other kinds of concepts that allow operators for extra flexibility in optimizing revenue. A study of market showcases is discussed by Kölle et al. (2020); Eguinoa et al. (2021). Furthermore, interest in load reductions, providing grid services, and more may become of interest to industry. For any new control strategy application, it will be necessary to follow these R&D processes from simulation to scaled testing to field demonstration.

## 5  Integrated design and systems perspective

As with other technological systems, the up-front design and the resulting control strategies that are available are not independent. Improved wind-farm flow control can inform design processes and affect design heuristics. For instance, the ability to increasing energy extraction and reduce loads, may enable the ability to build wind farms with increased power density (with higher AEPs per square kilometre).

Integrated design of the system hardware and software (or controls) is an emerging research area in wind energy with the potential to enable significant innovation (Garcia-Sanz, 2019). Control co-design (CCD), which focuses on concurrent design of all relevant disciplines, including controls, has been recognized as a promising research and technology development pathway for wind turbines and floating wind turbines in particular (Garcia-Sanz, 2019). However, the community also recognizes the need to look at holistic design and control of full wind farms due to the significant couplings of the flow and performance at the wind farm level. Thus, there is a need to elevate the application of CCD to the farm level. Essentially, since the location and types of wind turbines, sensing equipment, and other control-related technologies in the farm determine the availability and magnitude of the wind farm control opportunity, there is a coupling of the up-front design of the farm and the downstream execution of wind farm control. First, we briefly discuss the state-of-the-art in wind farm design before elaborating on recent efforts for integrated wind farm design and control.

### 5.1  Progress in wind farm design optimization research

Research in pre-construction wind farm design spans multiple decades (Herbert-Acero et al., 2014). As discussed in § 1.1, there are a number of objectives of interest for operation and control, including energy production and cost of energy. These same metrics are used in the pre-construction wind farm design process. Historically, AEP was the main objective and cost





elements have been increasingly incorporated to enable full LCoE (as a metric to represent overall farm economic profitability) (Ning et al., 2019).

As discussed in Dykes et al. (2021), the key ingredients of wind farm multidisciplinary design optimization correspond to:

- *Annual Energy Production*: In addition to the size (e.g., rotor diameter) and performance features (e.g., rated power) of the turbines, AEP of a wind farm is also characterized by their interactions through wake effects. With the objective of maximizing AEP, the turbine selection, the number of turbines, and the layout have all been considered in the optimization problem.

- *Capital Expenditure for the balance of systems*: CAPEX for BOS optimization is frequently considered as a key cost component within the suboptimization problem of the electrical/power system (Perez-Moreno et al., 2018). Alternatively, its own detailed optimization is performed once a layout has been decided (Pérez-Rúa et al., 2020). Other costs for BOS such as the roads for land-based farms, including the installation and logistics strategies have also been explored.

- *CAPEX for turbines and foundations*: Recent research has investigated coupled optimization of wind farm and turbine
design/selection (Stanley et al., 2018, 2019; Stanley and Ning, 2019; Graf et al., 2016) as well as coupling with design of offshore support structures (Perez-Moreno et al., 2018).

- *Operational Expenditure*: Originally, site suitability for loads was considered after a farm layout was mostly fixed. More recently, surrogate models has allowed inclusion of load models directly in wind farm the optimization (Riva et al., 2020).

For design, LCoE has served the wind energy community as a complex but straightforward metric, which allowed wind farm developers to largely ignore time-varying aspects of system performance and cost. Through statistical models, the time-varying nature of the wind resource was simplified into a wind rose and the annual energy production for a given design was calculated once per design iteration.

Looking towards future electricity systems with large levels of variable sources of energy generation and a shift toward
subsidy-free wind energy systems, the need to increase wind farm profitability will shift focus from LCoE towards objectives that also account for time-varying revenues and costs. Initial work in this direction has shown that the current level of generation of wind energy in a system can be parameterized to give an indication of the value of new wind generation as a function of statistical correlations between wind and market conditions (Simpson et al., 2020). Other metrics account for the time-varying revenue a wind farm would see over its lifetime when participating in merchant markets (Beiter et al., 2021). Such metrics
could also be used to evaluate the potential of control strategies as well as the co-design of the wind farm and control strategy together. This area of research is still largely unexplored but promises many new challenges for wind farm design, control, and the interaction of the two.

From a wind farm design perspective, there are various potential impacts of using value-based metrics over LCoE. We expect impacts on:

- Wind turbine design: A recent trend has explored the potential of low-wind-speed machines or machines with low specific-power that produce more electricity at lower wind speeds than conventional wind turbines (Simpson et al.,



2020). Such machines can be designed to "cut out" at lower wind speeds so that they only operate at lower wind speeds (reducing their LCoE compared to machines that operate over a larger wind speed range) (Madsen et al., 2020).


- Wind farm machine selection: Increasingly, wind farm designers can consider a spectrum of technology solutions as manufacturers move to platform models with a large number of variants including rotor diameter, rated power, hub height, and other features. Wind farm designers can select between these variants or some combination of them. With new low-wind turbines as in Madsen et al. (2020), it could be possible to create a wind farm that includes both conventional and low-wind machines with revenue under a broader range of resource and market conditions for higher overall system value.


- Wind farm meta-design: A key design lever for value maximization is the ability to constrain the capacity of the wind farm relative to the sum of the capacity of individual assets. This is done by increasing the number of turbines relative to the rated output of the facility (a strategy known as overplanting) (Dykes et al., 2019). This can also be accomplished through hybridization with other generation technologies (such as solar photovoltaics) or combining with storage technologies (Dykes et al., 2020).


- Wind farm physical design: This addresses the overall physical layout/placement of machines and the BOS design.

Physical design is difficult to separate from both topics of machine selection and meta-design in the process, due to significant couplings. The type and number of turbines, coupled with their placement in the farm, will influence the overall energy production of the farm. Also influencing the energy production is the overall operation of the plant and thus the control of the wind farm or hybrid power plant over its lifetime. Thus, holistic optimization of a wind farm necessarily brings together farm

design and control. Furthermore, the introduction of control strategies into the wind farm design problem, as will be discussed, can impact not just the wind farm performance but also trade-offs in system performance and cost.

## 5.2 Wind farm control co-design (CCD)

As previously mentioned, CCD is a promising field for wind energy research in general due to the strong couplings between the physical system design and the design of the software control system (Garcia-Sanz, 2019).

### 5.2.1 Wind farm CCD for AEP and LCoE objectives

Some of the first works looking at the subject for wind farm applications focused on static yaw-based wake steering control and wind farm layout optimization. In Fleming et al. (2016), combined wake steering and layout optimization was shown to increase the energy production of a wind farm at the same time as the farm power density was increased (moving the turbines closer together and thus enabling cost reductions in the balance of systems). Gebraad et al. (2017) extended the work by

considering a full wind rose of wind speed and direction to optimize the plant AEP. More recently, combined static induction control and layout optimization also demonstrated some potential for improving farm AEP (Pedersen and Larsen, 2020).

The above work did not address the loading conditions as influenced by either the turbine placement, control strategy, or both. Recent work by Riva et al. (2020) and Stanley et al. (2020) looked at the coupling of the layout and wind turbine loading under




normal operation. A key challenge for research in wind farm design and control optimization applications considering turbine
loading is the computational costs, as a single optimization iteration can become computationally intensive — even when
parallelizing across operational cases. This becomes even more challenging when considering CCD of the control strategy and
farm design together.

The opportunities and challenges for CCD become larger as the operating degrees of freedom of the system increase. Thus,
floating wind energy —which involves potential motions of the platform and turbine system in response to wind and wave
conditions— creates a significant opportunity and challenge for CCD research (Garcia-Sanz, 2019). First, floating wind energy
may be investigated similar to fixed-bottom or land-based wind farms for CCD to benefit AEP, reduce LCoE, or address other
objectives. However, floating wind energy may introduce additional design levers from the platform motion that might augment
the benefits that can be realized from CCD (Barter et al., 2020). Furthermore, recent research has looked at the ability of using
yaw-based control to displace floating wind turbines thus inducing not just wake steering but also shifting the entire turbine
and trajectory of the wake (Kheirabadi and Nagamune, 2020). To realize such a strategy, floating wind farm design would need
to take into account these displacements in the broader optimization of not just the nominal layout of the project but also the
topology and sizing of system components such as mooring systems, anchors, and the broader collection system.

### 5.2.2  Wind farm CCD for profitability objectives

Inherent in the entire discussion about wind farm design for value-based objectives beyond LCoE is an active control strategy
that can no longer be treated as independent from the farm design itself. Similarly, the design and operation of wind farms
for objectives beyond LCoE introduce further couplings of physical design and controls. By definition, overplanting means
that some or many wind turbines in a wind farm will not be operating or operating at derated conditions, depending on the
particular operational/control strategy. Similarly, hybridization of wind farms, including other generation assets, means that
there are many objectives to balance: the overall farm revenue against the overall selection and sizing of the different assets,
the physical park design, and the long-term reliability the facility, and minimizing the operational costs for the collection of
assets.

In an LCoE-driven world, the long-term reliability of the assets is important. But, with discount factors and tax incentives
favoring production in the early years of the project life cycle, designers and operators have focused largely on producing
maximum energy at the lowest possible cost. Introducing value-based objectives where there is an explicit trade-off in the
performance and reliability across the generation assets in a project necessitates a new approach to optimization that addresses
the operation and control strategy up-front in the design process. Figure 8 illustrates this trade-off in a simplistic way. In
short, the design of the system affects the operation and control strategies available, which in turn affect the overall project
profitability (balancing revenues against costs). By taking into account the control strategy in the design process, through
CCD, we can improve the life cycle profitability of the project. For wind energy systems with significant couplings between
individual assets, due to wakes and their impacts both on downstream machine power production and loading, CCD will be
necessary to realizing profitable wind farms in future highly variable energy systems.





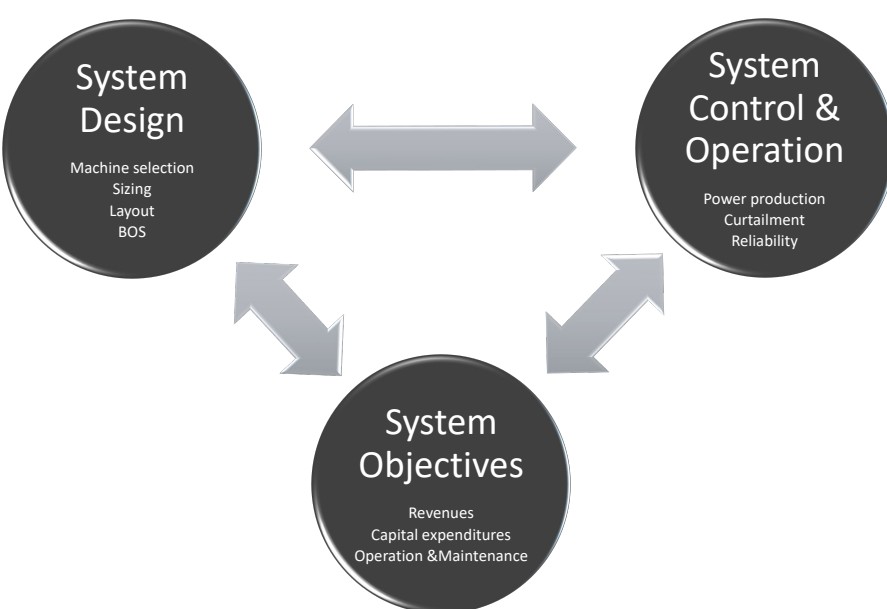

**Figure 8.** Illustration of interdependencies for CCD considering system design, control strategies and objectives.

### 5.2.3 Wind farm CCD for non-economic objectives

As already discussed, mitigation of environmental and/or social impacts can be an important objective of wind farm control. As with economic metrics, these are important for wind farm design and CCD as well. Current practice considers the external impacts of the wind farm design and operation/control in isolation. But there may be potential to further improve the overall farm economic performance while alleviating negative externalities through employment of CCD. One recent example of this is through the mitigation of wind farm noise impacts (Cao et al., 2020). Noise generation and propagation are affected by the design of individual turbines, their placement in a farm relative to each other and a potential impact point, as well as how the turbines are operated throughout their lifetime. While Cao et al. (2020) looked only at the layout influence on noise, allowing for varying operational strategies for noise along with the design optimization holds promise for further improvement of the economic performance of the farm while minimizing noise impacts. Other externalities to the environment and communities could be similarly addressed in CCD research.



# 6    Conclusions

In the current paper, we have identified four major areas in which important open scientific challenges and opportunities exist
for wind farm control: 1) wind farm flow control physics, 2) algorithms and AI, 3) validation and implementation, and 4)
integrating control with system design (co-design).

With respect to the first area (wind farm flow control physics), we distinguish between quasi-steady and dynamic control.
Quasi-steady induction control has been extensively studied, but remaining opportunities relate to its use for load reduction
and the use of overinduction for increasing energy extraction. In the field of quasi-static yaw control, further insights in wake
shape, and its combination with induction control are interesting open research topics. Overall, the effect of shear, veer, and
atmospheric stability on wake shape are also still poorly understood. Dynamic control is still at its infancy, with promising first
results in the field of individual pitch control and dynamic induction control, both for wake breakup. Whether it is possible
to control turbulence at a larger boundary-layer scale for improved mixing and energy extraction, also remains an interesting
fundamental research question. Finally, larger mesoscale effects of wind farms, such as blockage, gravity waves, wind farm
wakes, etc. may also be susceptible to control action. This area is fully unexplored to date and may lead to a lot of opportunities
for large wind farms.

In the second area (algorithms and AI), we highlight the opportunities and challenges related to a shift from open-loop
control (the current standard, e.g., mostly based on look-up tables), to closed-loop control as a means to reduce uncertainty
and reduce model errors. This becomes particularly important when considering more advanced control approaches in higher-
dimensional control spaces. Challenges relate to the choice and construction of appropriate control models (the internal model),
the necessary state estimation techniques to construct reliable virtual twins of turbines and flow, and possible trade-offs between
observability and type and number of sensors that are required for state estimation of more complex models. We foresee that
current progress in machine learning and artificial intelligence will become a key enabler to solve some of these outstanding
problems.

When considering the third area (validation and implementation), we highlight in particular the challenges associated to
testing and implementing new control in modern wind farms, with investment costs in the billion-euro range. Direct testing of
new ideas on the full scale is simply not possible, and instead a careful proof-of-concept and validation strategy is required.
To this end, we foresee that in the coming years, the sequence of large-eddy simulations, wind tunnel experiments, and small
field campaigns will play an ever larger role. Each of these faces its own challenges, respectively related to model bias, scale
similarity, and establishing statistical significance. Moreover, for actual commercial implementation, additional issues arise,
such as controller safety and proof of commercial value for different sites, among others.

Finally, we highlight a fourth area (co-design), which we believe to become instrumental in reaching the full potential of
wind farm control. With the evolution towards subsidy-free wind energy and more variable energy markets, objectives for
wind farm design (and control) are moving from LCOE to value-based metrics that incorporate the time-varying nature of
energy prices as well as costs. Incorporating wind farm control in the design optimization process will lead to denser farms
that optimally exploit varying conditions. During high electricity prices, control can be used to mitigate wake effects, while



during low prices, more emphasis can be given to load reduction and lifetime extension. Moreover, in either scenario, ancillary service provision may provide an alternative income base. For instance, in this context, it is interesting to note that the Belgian government recently increased the capacity of the new Princess Elisabeth offshore wind development zone with 60% (from

2.2 GW to 3.5 GW), based on the expectation of a shift of the techno-economic optimum as a result of technological progress (O'Brian, 2021).

In summary, the field of wind farm flow control is an active area of research and innovation, with many interesting multidisciplinary challenges, and exciting prospects for the increase of the total value of wind energy for society.

**Nomenclature**

AEP     Annual Energy Production

AI      Artificial Intelligence

BOS     Balance of System

CAPEX   Capital Expenditure

CCD     Control Co-Design

CFD     Computational Fluid Dynamics

DMD     Dynamic Mode Decomposition

HPC     High Performance Computing

LCoE    Levelised Cost of Energy

LES     Large Eddy Simulations

LUT     Look-Up-Table

O&M     Operations & Maintenance

OPEX    Operational Expenditure

PIV     Particle Image Velocimetry

R&D     Research and Development

TRL     Technology Readiness Level

TSO     Transmission System Operator



WFFC   Wind Farm Flow Control

*Author contributions.* Conceptualization (JM, CB, KD, PF, PG, GG, TG, JWvW); Writing — Original Draft (JM, CB, KD, PF, PG, TG,
JWvW); Writing — Review  Editing (JM, CB, KD, PF, PG, GG, TG, JWvW)

*Competing interests.* The authors have the following competing interests: Johan Meyers, Carlo Bottasso and Katherine Dykes are members
of the editorial board of Wind Energy Science. The peer-review process was guided by an independent editor, and the authors have no other
competing interests to declare.

*Acknowledgements.* The authors express their gratitude to Dr. Bart Doekemeijer, Dr. Wim Munters, and Dr. Vlaho Petrović for their useful
comments and feedback on this manuscript.

JM, CB, KD, GG, TG, and J-WvW acknowledge support from the FarmConners project, funded by the European Union's Horizon 2020
research and innovation programme under grant agreement No 857844.

This work was authored [in part] by the National Renewable Energy Laboratory, operated by Alliance for Sustainable Energy, LLC, for
the U.S. Department of Energy (DOE) under Contract No. DE-AC36-08GO28308. Funding provided by the U.S. Department of Energy
Office of Energy Efficiency and Renewable Energy Wind Energy Technologies Office. The views expressed in the article do not necessarily
represent the views of the DOE or the U.S. Government. The U.S. Government retains and the publisher, by accepting the article for
publication, acknowledges that the U.S. Government retains a nonexclusive, paid-up, irrevocable, worldwide license to publish or reproduce
the published form of this work, or allow others to do so, for U.S. Government purposes.





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
