# Peer review of "Wind farm flow control: prospects and challenges"

_Wind Energy Science, 2022_

## Author Comment (AC1)

**Wind farm flow control: prospects and challenges — Reply to Reviewer 1**

**Reviewer:** *A very thorough and comprehensive review of the current status of wind farm flow control. Some sections are perhaps a bit wordy, and there will always be scope to include a few more references. Just a very few small and specific comments:*

**Response:** We thank the reviewer for his/her very supportive comments, and have readily addressed the issues raised by the reviewer in the revised manuscript (see below for details)

1. **Reviewer:** *Line 142-3: "While these types of precise implementation details can matter, e.g., for turbine loading" - these details can be more significant than this implies; the exact means by which a particular wake change is achieved can affect power output as well as loads.*

   **Response:** Thank you, this is indeed a good point. We have improved the discussion on this part as follows

   "However, when discussing wind farm control, very often collective effects on the flow physics that result from turbine actuation are straightaway considered as a control input, without directly considering the precise actuation at the turbine level. The most common example is induction control, in which the axial induction set-point of the turbine is changed to affect the wake and its downstream interactions. This may be achieved in various ways, i.e. by changing the generator-torque set point (thus changing the  tip-speed ratio), the collective blade pitch angles, or combinations thereof. We should note that, although these details do not matter much for the effective wake-flow development, they do matter in terms of loads and power, and should be included in the overall control optimization. For instance, derating the turbine without pitching the blades is sub optimal in terms of power extraction, i.e. given a thrust set-point, there is a unique pitch–tip-speed-ratio combination that maximizes power. When considering yaw control, changes in the yaw set-point can lead to changes in the trust set-point as well, which need to be properly captured for correct wake behavior. Again, these changes can include changes in generator-torque or blade-pitch set-points, and precise details can matter a lot for the effective power output and turbine loads. Finally, we note that, given a selected thrust and yaw set point, the effective turbine torque (and related power set-point) will have a subtle effect on the amount of wake rotation induced by the turbine, but these effects are small, given that modern turbines operate at high tip speed ratios."

2. **Reviewer:** *Line 182: wake steering combined with overinductive induction control: this could further exacerbate loading; whereas the option to combine wake steering with 'normal' induction control (not necessarily simultaneously on a particular turbine but depending on turbine position and wind condition) should also me mentioned (and referenced) in this paragraph - it's one option for achieving a suitable compromise between energy production and loading.*

   **Response:** This is a good point. We changed the discussion in the

manuscript as follows

" Yawing a turbine to redirect its wake, increases the turbine loading. Combining yaw control with derating (underinduction) of the turbine can be used to find a trade off between energy extraction and load reduction at the level of the farm (Bossanyi, 2018; Debusscher et al., 2022). Another track that has received some attention is the combination of yaw control and overinduction. ... "

3. **Reviewer:** *Line 210: "resort under" - do you mean "result in", or something implying "be equivalent to"?*
   **Response:** We changed the formulation into "an approach that could technically be categorized as 'static' wake redirection "

4. **Reviewer:** *Line 444: typo "leveraged"*
   **Response:** Thank you, has been corrected.

5. **Reviewer:** *Line 950: "reliability of the facility"*
   **Response:** Corrected.

---

## Author Comment (AC2)

**Wind farm flow control: prospects and challenges — Reply to Reviewer 2**

**Reviewer:** *This paper presents a comprehensive overview of wind farm flow control, its current status, as well as challenges for its practical demonstration and commercialization, organized under four key research areas. Although it's always difficult to find a balance between length and depth for such a wide technological field, authors provide an excellent coverage of the most relevant aspects discussed within the community and an extensive list of references.*

**Response:** We thank the reviewer for her detailed comments, and have readily addressed the issues raised by the reviewer in the revised manuscript (see below for details)

**Specific comments**

1. **Reviewer:** *Lines-87-88: The sentence "Still, some clear benefits. . . " could be supported with a reference to section 5.1, where those aspects are further discussed.*
   **Response:** Thank you for the suggestion. We have added a reference to Section 5.1 in the manuscript.

2. **Reviewer:** *The way of presenting section 2.1, dealing with static wake control aspects, seems to mismatch the approach for the rest of section 2, where aspects of the flow physics are analyzed instead. From section 2.1, it could be inferred that the challenges presented only affect the static controls rather than being related to the (quasi)-steady dynamics of the flow physics (as mentioned in Line-228). Could you please clarify?*
   **Response:** We realize that the naming of the sections may be misinterpreted (steady and dynamic flow physics in the wake rather then quasi-steady and dynamic control), so we changed them to "2.1 Quasi-steady flow control physics", and "2.2 Dynamic flow control physics". Thus, to our understanding/definition, the terms 'quasi-static' and 'dynamic' are opposites, and so (quasi)-steady dynamics of the flow physics does not make sense.

   Moreover, to further clarify, we added as well following additional explanation in the introductory description of 2.1:
   "... Over the years, this type of control has been extensively studied, so that the response of the wake to control actions is relatively well documented. Nevertheless, open questions remain when considering, e.g. near-wake behaviour, impact of atmospheric conditions, effects of wake shape and deficit on loads, etc., as further discussed in §2.1.1 for steady axial induction control and in §2.1.2 for steady yaw control. "

3. **Reviewer:** *At the end of section 2.2.2, in Line-372, it is stated that the use of LES simulations as a control model is hindered by the large cost and complexity associated. Then, further in the text (Line-382), a related discussion is raised about the relevant factors (minimum resolution, model simplifications) when considering its potential application in real time. Could those factors also be applicable to the offline case initial discussion?*

**Response:** Absolutely, but maybe it is even more correct to state that the work of Goit and later Munters is not true optimal control, but rather optimization of controls given perfect knowledge of the system. Translating the approach into a real optimal control framework would yield the challenges discussed at the end of 2.2.2 (line 382 in the original manuscript). Therefore, we lightly reformulated the first statement into:

"However, from a control implementation point of view, this approach is hindered by the large cost associated with large-eddy simulations as a control model. Moreover, strong simplifying assumptions were used, i.e. perfect knowledge of the state, and a control model that exactly matches the (virtual) plant (Munters and Meyers, 2017). Unfortunately, ... "

4. **Reviewer:** *Line-483: Could you please specify/clarify the exact meaning of "cautious decisions" in this context?*
   **Response:** It is to be read in combination with the uncertainty assessment embedded in the decision making process, thus, the cautious decisions refer to conservative actions and/or safety margin inclusion. Now referred in the text as "In combination with an uncertainty description, it will be used to make cautious (with a safety margin) or robust decisions about the control settings of the individual turbines in a receding horizon framework, ..."

5. **Reviewer:** *Section 3.2 — The overall challenges for the closed-loop paradigm are mentioned in Lines-488-489, but the rest of the section seems to be at some parts just a description of the state of the art rather than an identification and further development of the corresponding challenges (e.g. state estimation paragraph). It would be advised to clearly identify the specific challenges addressed.*
   **Response:** Thank you; this was indeed the case. We have reformulated and moved parts in the section, to better streamline the text and highlight the challenges.

6. **Reviewer:** *Section 3.2 — Novel optimization routines are identified as a challenge for the closed-loop paradigm (Lines-488-489), but this isn't truly developed in the corresponding paragraph devoted to "robust decision making". Some aspects that are important are listed below, but it is unclear whether authors considered all those aspects as challenges (unresolved issues) or just relevant factors in the selection of the optimization algorithm.*
   **Response:** Now the paragraph is rephrased further to underline the challenges identified clearly.

7. **Reviewer:** *Section 3.3: For the sake of clarity, could you please explain in more detail in what data-driven workflows (Line-563) consists of as opposed to physics-based workflows (Line-563) and AI-driven workflows (Line-579)? Maybe a diagram or short description would be of help to make the distinction.*
   **Response:** In the article, the idea is to discuss the synergies in summary instead of many details focusing on AI for wind energy or WFFC. Therefore, the useful explanations pointed out by the reviewer are kept briefly in the Section 3.3 text (in parenthesis). Data-driven workflow refers to building the system based on observations, where the decisions are typically

made by humans (humans as the processors). Physics-based workflows build the system based on physical representations and/or set of rules and the decisions are also made typically by humans. For the AI-driven workflows, the system can be built up via the observations and/or physical representations (examples of such hybrid approaches are also cited in the Section) but crucially, the decision making process is also automatized and AI is the central processor.

8. **Reviewer:** *Section 4.4. Does the section only apply to WFFC technology developments performed by OEMs or is it extensive to any other technology provider? If the latter is the case, shouldn't it be considered as a relevant challenge the (standardized) access to information between farm and turbine level and the communication interface of WFFC with turbine control?*

   **Response:** The section describes the steps to follow in R&D processes from proof of concept to commercialization, in order to develop a reliable, safe, and scalable wind farm control product. The steps described may be followed by any party or combination of parties. Indeed, standardization, in a more general sense rather than only referring to the interfaces, helps for several parties to work together on a solution, and for developing solutions faster. We have added a short mentioning of this at the end of the section: " Moreover, to accelerate the development of new wind farm control products, the industry could benefit from standardization of the processes, and of the measurement and controls interfaces, such as, for instance the communications interface between WFFC and turbine controller." Standardization of practices may also be to the benefit of manufacturers in working with other suppliers, wind park developers, and certifiers, for example.

**Technical corrections**

1. **Reviewer:** *Lines-56-57. Reference to Figure 1 seems to be a bit out of place. The content of the figure seems to be more in relation with the discussion in Section 1.2 rather than that at the beginning of Section 1, where the figure is introduced in the text (Lines-56-57).*

   **Response:** We agree that the figure is mostly illustrating some of the control physics, and definitely not all aspects of WFFC. Nevertheless, we prefer to keep the figure in the introduction, but have reorganized the text a bit to better match the content of the figure. On pate xx, line xx, we now describe: "The focus of this manuscript is on wind farm flow control. We define it as the coordinated control of the turbines in the farm with the aim to influence the flow (wakes, turbulence) in such a way that it improves the overall figure of merit of the farm. The latter can be, e.g., overall power extraction, total lifetime, levelized cost of energy, or simply the lifetime profit. A graphical impression of WFFC is provided in Figure 1, highlighting in particular some of the physics that can be leveraged for influencing the flow (see also §1.2 for more details). Note that ... "

2. **Reviewer:** *Taking into account the different meanings of the term "loading" depending on the discipline, it would be advised to clarify that unless*

*specified otherwise, it refers to structural loading. First use in Line-18.*
**Response:** Good point - now the 'load' or 'loading' is re-worded as 'structural load' or 'structural loading' where applicable throughout the article.

3. **Reviewer:** *The acronym for wind farm flow control (WFFC) is defined in Line-36, but it scarcely appears afterwards throughout the paper despite being one of the most mentioned terms. Authors are encouraged to make use of it in order to lighten the text. Please also note that its first appearance is in Line-33.*
**Response:** We have mostly replaced 'wind farm flow control' with 'WFFC' as suggested by the reviewer (not marked with track changes in the revision)

4. **Reviewer:** *Line-380: Is it meant to say "real-time controller" instead of "real controller"?*
**Response:** Yes, indeed - typo corrected now.

5. **Reviewer:** *Caption Figure 5 – For the sake of clarity, please specify the type of control scheme depicted (open-loop), in accordance with the explanation in the text.*
**Reviewer:** *Caption Figure 6 – For the sake of clarity, please specify the type of control scheme depicted (closed-loop), in accordance with the explanation in the text.*
**Response:** Thank you for these suggestions; implemented.

6. **Reviewer:** *Acronyms – LES -¿ first instance in the text is in Line-170 instead of Line-180.*
**Response:** thank you for pointing this out. Moved definition of acronym accordingly.

7. **Reviewer:** *Line-506: typo "uses techniques"*
**Response:** corrected

8. **Reviewer:** *References: Please try to make all references discoverable with either DOI link (if applicable) or direct access link. Reference in Line-1260: is the author properly presented? Some references are missing the publication year: Line-1380, Line-1384.*
**Response:** Thank you. We have corrected the references. With respect to DOIs and direct access links — this is a good point; once the manuscript is accepted, we will work with Copernicus during proofreading to include these as much as possible.

---

## Author Response (AR2)

**As requested, we have implemented following corrections**

The title page of *pdf. manuscript file must include the full institutional addresses of all authors. However, country name is missing from the affiliation #3. Please add it for the next revision

⇨ DONE

I have just found a small typo on Line 149 of the revised manuscript (pdf with track changes) : trust instead of thrust. since the word exists, I am afraid that if we do not correct it now, it might slip through the copy-editing process.

⇨ Thanks for noticing – DONE